# The structure and function of the global citrus rhizosphere microbiome

Jin Xu [1,2], Yunzeng Zhang [1,22], Pengfan Zhang [3,4,5], Pankaj Trivedi[6], Nadia Riera[1], Yayu Wang[3,4], Xin Liu [3,4,7], Guangyi Fan[3,4,7], Jiliang Tang[8], Helvécio D. Coletta-Filho[9], Jaime Cubero [10], Xiaoling Deng[11], Veronica Ancona[12], Zhanjun Lu[13], Balian Zhong[13], M. Caroline Roper[14], Nieves Capote[15], Vittoria Catara[16], Gerhard Pietersen[17], Christian Vernière[18,19], Abdullah M. Al-Sadi[20], Lei Li[1], Fan Yang[4], Xun Xu[3,4,7], Jian Wang[3,4], Huanming Yang[3,4], Tao Jin[3,4,7] & Nian Wang [1,21]

Citrus is a globally important, perennial fruit crop whose rhizosphere microbiome is thought to play an important role in promoting citrus growth and health. Here, we report a comprehensive analysis of the structural and functional composition of the citrus rhizosphere microbiome. We use both amplicon and deep shotgun metagenomic sequencing of bulk soil and rhizosphere samples collected across distinct biogeographical regions from six continents. Predominant taxa include *Proteobacteria*, *Actinobacteria*, *Acidobacteria* and *Bacteroidetes*. The core citrus rhizosphere microbiome comprises *Pseudomonas*, *Agrobacterium*, *Cupriavidus*, *Bradyrhizobium*, *Rhizobium*, *Mesorhizobium*, *Burkholderia*, *Cellvibrio*, *Sphingomonas*, *Variovorax* and *Paraburkholderia*, some of which are potential plant beneficial microbes. We also identify over-represented microbial functional traits mediating plant-microbe and microbe-microbe interactions, nutrition acquisition and plant growth promotion in citrus rhizosphere. The results provide valuable information to guide microbial isolation and culturing and, potentially, to harness the power of the microbiome to improve plant production and health.

[1] Citrus Research and Education Center, Department of Microbiology and Cell Science, Institute of Food and Agricultural Sciences (IFAS), University of Florida, Lake Alfred 33885 FL, USA. [2] Citrus Research and Education Center, Department of Plant Pathology, IFAS, University of Florida, Lake Alfred 33885 FL, USA. [3] BGI-Shenzhen, Shenzhen 518083 Guangdong, China. [4] China National GeneBank, BGI-Shenzhen, Shenzhen 518083 Guangdong, China. [5] BGI Education Center, University of Chinese Academy of Sciences, Shenzhen 518083 Guangdong, China. [6] Department of Bioagricultural Sciences and Pest Management, Colorado State University, Fort Collins 80523 CO, USA. [7] BGI-Qingdao, BGI-Shenzhen, Qingdao 266510 Shangdong, China. [8] Guangxi University, Nanning 530004 Guangxi, China. [9] Instituto Agronômico, IAC Centro de Citricultura Sylvio Moreira, CCSM, Cordeirópolis 13490 São Paulo, Brazil. [10] Dept. Plant Protection, Instituto Nacional de Investigación y Tecnología Agraria y Alimentaria (INIA), Madrid 28040, Spain. [11] Department of Plant Pathology, South China Agricultural University, Guangzhou 510642, China. [12] Texas A&M University-Kingsville Citrus Center, Weslaco 78599 TX, USA. [13] National Navel Orange Engineering Research Center, Gannan Normal University, Ganzhou 341000 Jiangxi, China. [14] University of California, Riverside 92521 CA, USA. [15] IFAPA Las Torres, Sevilla 41200, Spain. [16] Dipartimento di Agricoltura, Alimentazione e Ambiente, University of Catania, Via Santa Sofia 100, 95123 Catania, Italy. [17] Department of Genetics, University of Stellenbosch, 7600 Stellenbosch, South Africa. [18] CIRAD, UMR BGPI, F-34398 Montpellier, Hérault, France. [19] CIRAD, UMR PVBMT, F-97410 St Pierre, La Réunion, France. [20] Department of Crop Sciences, Sultan Qaboos University, Muscat 123, Oman. [21] China-USA Citrus Huanglongbing Joint Laboratory (A joint laboratory of The University of Florida's Institute of Food and Agricultural Sciences and Gannan Normal University), National Navel Orange Engineering Research Center, Gannan Normal University, Ganzhou 341000 Jiangxi, China. [22] Present address: College of Bioscience and Biotechnology, Yangzhou University, Yangzhou 225009 Jiangsu, China. These authors contributed equally: Jin Xu, Yunzeng Zhang, Pengfan Zhang.  Correspondence and requests for materials should be addressed to T.J. (email: jintao@genomics.cn) or to N.W. (email: nianwang@ufl.edu)

The rhizosphere harbors diverse microbes, many of which undoubtedly benefit plants by preventing pathogenic infection and assisting in nutritional acquisition from the soil. Understanding the taxonomic, genomic, and functional components of the rhizosphere microbiome is crucial for managing them for sustainable crop production[1,2]. Progress has been made toward the characterization of rhizosphere microbiomes in certain model and crop plant species, including *Arabidopsis*[3,4], rice[5], millet[6], soybean[7], corn[8], barley[9], wheat[10], sugarcane[11], cucumber[12], citrus[13], populus[14], and grapevine[15] by exploring the structure, functional genes, and factors that determine assembly of the microbiome. Most studies of plant-associated microbial communities have been conducted by means of ribosomal amplicon-based approaches[3–6,8,10,11,14–16]. However, amplicon-based community profiling does not provide either the genomic or functional details of the microbiome. Research priorities for harnessing plant microbiomes for sustainable agriculture include determining the functional mechanisms mediating plant–microbiome interactions and defining the core microbiome of crop and non-crop plant species[2]. These research priorities require metagenomic studies of the microbiome that can provide taxonomic, genomic, and functional information for a given community. Such whole-genome shotgun sequencing-based metagenomic studies have been conducted for human, animal, and oceanic microbiomes[17,18] as well as complex soil communities[9,12,13,19–22]. However, the global pattern of the genomic and functional contents of rhizosphere microbial communities remains largely unexplored; such information is needed to understand and manage microbial functions in agroecosystems in support of enhancing global agriculture in the future[23].

Citrus originated in southeast Asia[24–26], and cultivation of citrus began at least 4000 years ago[26]. Citrus is a large genus including several major cultivated species. Citrus is cultivated in more than 140 countries that define several distinct biogeographical regions. The total worldwide citrus acreage was approximately 9 million hectares with a production of 122.3 million tons in 2009, making citrus the largest fruit crop[27]. The genome sequences of several citrus germplasms have been deciphered[27–30], extending the capacity to effectively genetically interrogate plant–microbiome interactions. Citrus thus represents an ideal model system to study the taxonomic, genomic, and functional components of the rhizosphere microbiome at a global scale. Moreover, citrus production worldwide has recently been hampered by environmental and disease pressures[31], and harnessing citrus–microbiome interactions to address biotic and abiotic stresses offers an opportunity to increase sustainable citrus production.

The International Citrus Microbiome Consortium was established in 2015[32]. Members of the consortium sampled rhizosphere and bulk soil samples from six continents representing distinct biogeographical regions, and performed amplicon and deep shotgun metagenomic sequencing of the rhizosphere and bulk soil samples to define the rhizosphere in each setting. Here, we present the results of this comparative study and define the genomic and functional features of the citrus rhizosphere from metagenomic sequencing of the community members, laying a foundation for harnessing the microbiome for sustainable citrus production.

## Results

**Taxonomic features of the global citrus rhizosphere microbiome.** Citrus rhizosphere and the associated bulk soil samples were collected from 23 representative locations in eight major citrus-producing countries spanning all six continents where citrus is grown[33] to explore the identity of the microbes constituting the rhizosphere microbiome and their genomic and thus functional features (Fig. 1, Supplementary Table 1, Supplementary Figure 1 and Supplementary Data 1). These 23 locations included 7 different soil types, and 6 climate types, with soil pH varying from pH 5.2 to 8.8, and with highly variable organic C, N, and P contents. A total of 12 citrus varieties were assessed. We

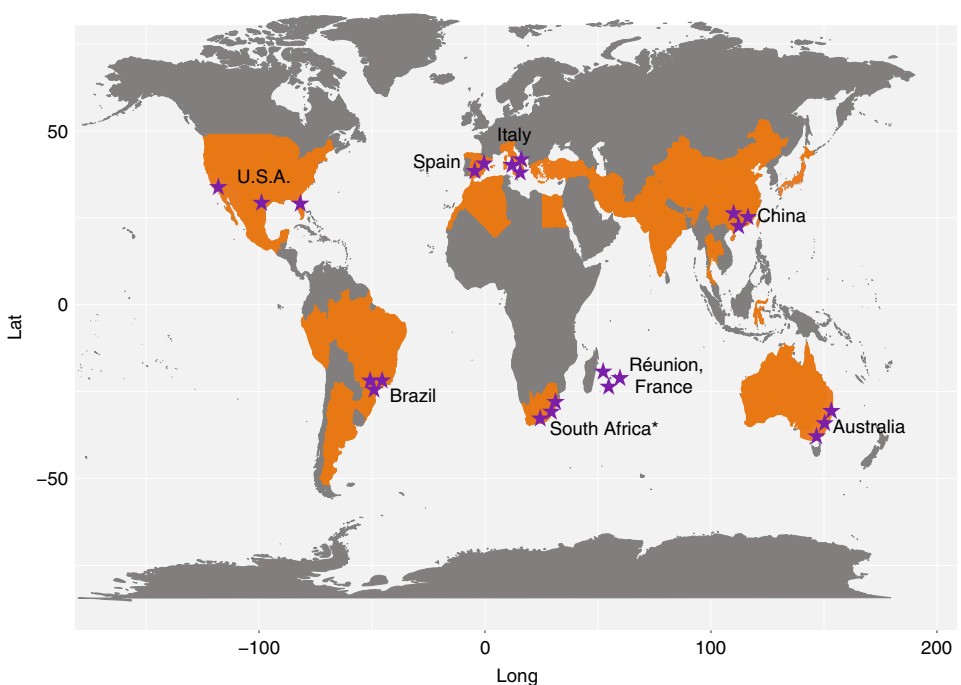

**Fig. 1** Geographic distribution of sampling sites across the world's citrus producing regions. Purple stars, the sampling sites; orange, major citrus producing countries. Map was adapted according to FAO data from 2016 (ref. [33]). Asterisk: only rhizosphere soil samples were collected

performed both amplicon (16S rDNA and ITS2) and deep shotgun metagenomic sequencing of both the rhizosphere and bulk soil samples. Approximately 1.22 and 0.76 million high-quality sequence tags were generated for the 16S and ITS sequencing samples, respectively (Supplementary Table 2, 3). After removal of sequences associated with the citrus host, on average 21,942 and 22,797 16S rDNA tags and 21,523 and 22,555 ITS2 tags were generated for each bulk soil and rhizosphere sample, respectively (Supplementary Table 2, 3).

More than 1.49 terabase pairs (Tbp) of shotgun metagenomic sequences were generated, yielding an average of 231.2 million (standard deviation, 3.55) paired-end reads (150 bp) for each sample. After the removal of sequences of citrus (on average, 1.16% of the clean reads) (Supplementary Table 4), de novo assembly was performed using a groupwise approach (see Methods, Supplementary Figure 2 and Supplementary Table 5). On average, 48.3% of the reads were utilized for metagenomic contig construction (Supplementary Table 5), a proportion that is higher than previous studies[12,13]. Approximately 230 million metagenes were predicted from the metagenomic contigs. These metagenes were then clustered into about 183 million non-redundant genes (unigenes).

Amplicon-based community composition analysis is a classical approach for microbiome analysis. However, shotgun metagenomic sequences generated without PCR amplification can also be used to determine the identity and relative abundance of microbes whose presence might not be detected in ribosomal gene amplicons due to primer bias[9] and have been successfully utilized in the interrogation of diverse microbiomes[9,12,17,18]. Here, we determined the identity of microbes in the citrus rhizosphere and associated bulk soil primarily by metagenomic sequencing, but complemented such assessments using amplicon sequences. Taxonomic annotations were assigned to 40.66% of the unigenes, 40.48% of which were prokaryotic (bacteria and archaea) (representing 99.55% of the total annotated unigenes). Only 0.17% of the unigenes were annotated as eukaryotic (including fungi, protozoa, algae, and plants) (representing 0.41% of the total annotated unigenes). Viral genes represented only 0.02% of the annotated unigenes, indicating potential annotation biases that lead to the underestimation of eukaryotic and viral communities. Such results support the conjecture that our understanding of soil/plant microbiomes is at an early stage, with little known about many community members[34]. Estimates of community composition made from amplicon and metagenomic sequences were highly concordant at the phylum level (pairwise Spearman's correlation coefficient $R^2 > 0.85$ for 16S and unigenes, and $R^2 > 0.73$ for ITS2 and unigenes). The dominant prokaryotic phyla found in the citrus rhizosphere included Proteobacteria, Actinobacteria, Acidobacteria, and Bacteroidetes, while fungal phyla, included Ascomycota and Basidiomycota (Fig. 2a and Supplementary Figure 3, 4).

We have previously observed the difference between the microbial diversity in the bulk soil and the rhizosphere of citrus using samples from the same location and observed the effect of root exudates on the composition of the rhizosphere community[35]. We investigated the taxonomic distinctiveness of the citrus rhizosphere and bulk soil microbiomes with samples across-the-globe here. No significant difference between the alpha diversity of the bulk soil and rhizosphere was seen; overall (P-value > 0.05, pairwise ANOVA) or within-site comparison (P-value > 0.05, one-sided t-tests, Fig. 2b and Supplementary Figure 5a). Samples from Brazil were the exception, however, with rhizosphere and bulk soils differing in community composition (P-value < 0.05 based on t-tests). Principal coordinate analysis (PCoA) and variation partitioning analysis (VPA) based on unweighted UniFrac distance (beta diversity) also

revealed that the community composition of the rhizosphere and bulk soil did not differ (P-value > 0.05, F-value = 1.23 using permutation-based ANOVA, Fig. 2c and Supplementary Figure 5b). We then compared the relative abundances of microbes in the bulk soil and rhizosphere at both high (phylum) and low (genus) taxonomic ranks using amplicon-based and metagenomic sequences to identify those community members differing in abundance in these two habitats (corrected P-value < 0.05, DESeq2, Supplementary Data 2–5 and Supplementary Figure 6, 7). Both amplicon-based and metagenomic approaches revealed that multiple bacterial phyla, such as Proteobacteria and Bacteroidetes, were present at a higher relative abundance in the rhizosphere, whereas multiple archaea phyla, such as Crenarchaeota, Euryarchaeota, and Thaumarchaeota were recovered at lower relative abundance in the rhizosphere (corrected P-value < 0.05, DESeq2, Supplementary Data 2 and 3). No fungal phyla exhibited any difference in relative abundance between rhizosphere and bulk soil (corrected P-value > 0.05, DESeq2, Supplementary Data 2 and 3). At lower taxonomic ranks, 142 genera were enriched in the rhizosphere microbiome, whereas 160 genera were depleted (corrected P-value < 0.05, DESeq2, Supplementary Data 4, 5 and Supplementary Figure 6, 7). The rhizosphere-enriched prokaryotic genera were over-represented in the phyla of Proteobacteria and Bacteroidetes (corrected P-value < 0.05, Fisher's exact test), whereas the depleted genera were mainly affiliated with Cyanobacteria, Firmicutes, Actinobacteria, and Euryarchaeota. Several genera distributed in Ascomycota were enriched in the rhizosphere microbiome (Supplementary Data 4, 5 and Supplementary Figure 7).

**Core taxa of the global citrus rhizosphere microbiome.** Since the metagenomic sequences obtained provided more comprehensive taxonomic information and given the community compositions made by this method were consistent with that from the amplicon sequences, this method was chosen to define a core rhizosphere microbiome. The metagenomic sequences also provided both taxonomic and functional information of a given taxon[9]. The core taxa of the rhizosphere microbiome were identified based on the following criteria: the genera were both enriched in the rhizosphere compared to the corresponding bulk soil samples and present in more than 75% of the samples from across-the-globe[6]. By such a metric, the majority of the identified microbes, namely, 1677 of 2193 (76.5%) genera were present in more than 75% of the rhizosphere samples and 138 genera were enriched from bulk soil to the rhizosphere. Using the aforementioned criteria, 132 genera were identified in the core citrus rhizosphere microbiome (Fig. 3a and Supplementary Data 5). The core rhizosphere microbes, such as Pseudomonas, Agrobacterium, Cupriavidus, Bradyrhizobium, Rhizobium, Shinella, Mesorhizobium, Burkholderia, Cellvibrio, Sphingomonas, Variovorax, Paraburkholderia, Dyadobacter, Novosphingobium, Devosia, and Ensifer, were over-represented in Proteobacteria (corrected P-value < 0.05, Fisher's exact test) (Fig. 3b and Supplementary Figure 7a). Multiple members affiliated with these core bacterial genera are known as plant beneficial microbes, and these microbes might help maintain plant hormone balance, control root development, facilitate nutrition acquisition, and prevent disease in the plant host[36,37]. In addition, seven core rhizosphere fungal genera, affiliated with Ascomycota, were identified as core microbiota (Fig. 3b and Supplementary Figure 7a). Some of these core fungal genera, such as nonpathogenic Fusarium and Hirsutella, may be potential biocontrol agents that could control fungal and nematode pathogens, respectively[38,39]. Exophiala and Colletotrichum have been suggested to promote

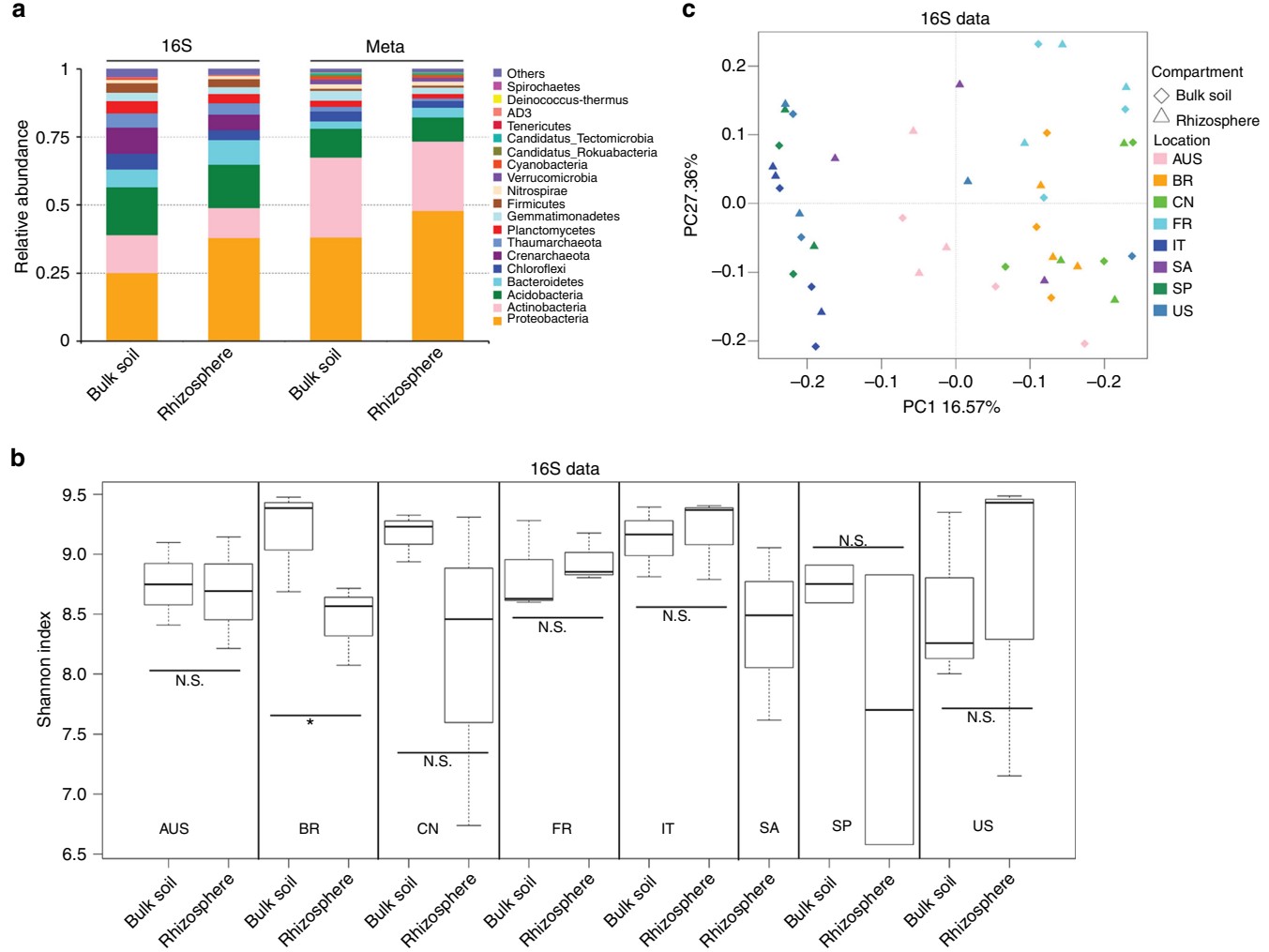

**Fig. 2** Taxonomic distribution and diversity comparisons in citrus rhizosphere and bulk soil microbiomes. **a** Phyla-level distributions in the bulk soil and rhizosphere samples based on 16S amplicon and metagenomic data. **b** Alpha diversity comparison between the bulk soil and rhizosphere samples from each location based on the Shannon index using the 16S data; N.S. no significant, *P-value < 0.05. One-sided *t*-test; center value represents the median of Shannon index. **c** PCoA based on the unweighted UniFrac distance between the bulk soil and rhizosphere for each location using the 16S data. AUS Australia, BR Brazil, CN China, FR French Réunion island, IT Italy, SA South Africa, SP Spain

plant growth via hormone production and phosphorus absorption, respectively, under abiotic stresses[40,41].

**Core functional traits of the global citrus rhizosphere microbiome.** The unigene set of the citrus rhizosphere and bulk soil microbiomes was four and 18-fold larger than that of the human gut and Tara oceanic microbiomes, respectively, and both human and Tara data were saturated[18,42]. However, rarefaction analysis of our unigene set indicated that the unigene sets for both citrus rhizosphere and bulk soil microbiomes did not reach a plateau (Fig. 4a), suggesting higher diversity and complexity within the citrus rhizosphere and bulk soil microbiomes remains to be found. Functional annotations were obtained for approximately 54% of the unigenes (98.8 of 183 million) by blasting against the KEGG Orthology (KO) database, and 15,610 KOs were assigned to the annotated unigenes. The KOs were mainly involved in 23 KEGG level 2 pathways (Fig. 4b). In total, 10,258 out of 15,405 (66.6%) and 10,749 out of 15,072 (71.3%) KOs were identified in at least 75% of the rhizosphere and bulk soil microbiomes, respectively, that were examined. A pairwise comparison of the rhizosphere and bulk soil samples revealed that 1748 KOs were

enriched in the rhizosphere samples, whereas 1417 KOs were depleted. Using such a 75% representation and enrichment as in the core taxa analysis, we defined the core and depleted functional traits of the citrus rhizosphere microbiome. Consequently, 1620 core and 1400 depleted functional traits were identified in the citrus rhizosphere (corrected *P*-value < 0.05, DESeq2, Fig. 4c, d and Supplementary Data 6). These core functional traits were mainly involved in plant–microbe and microbe–microbe interactions and pathways that might be involved in nutrient acquisition of microbes. The rhizosphere-depleted functional traits were involved in genetic information processing and metabolic pathways, such as carbohydrate metabolism, amino acid biosynthesis, energy metabolism, and nucleic acid biosynthesis (Fig. 4e, Supplementary Figure 8 and Supplementary Data 6).

Plant–microbe and microbe–microbe interactions are very likely to be important factors that would influence the assembly of rhizosphere microbiomes. The KOs involved in known plant–microbe and microbe–microbe interactions, such as bacterial secretion systems, flagella assembly, bacterial chemotaxis, bacterial toxins, bacterial motility, two-component system, and biofilm formation, were over-represented in the core rhizosphere microbiome (corrected *P*-value < 0.05, Fisher's exact

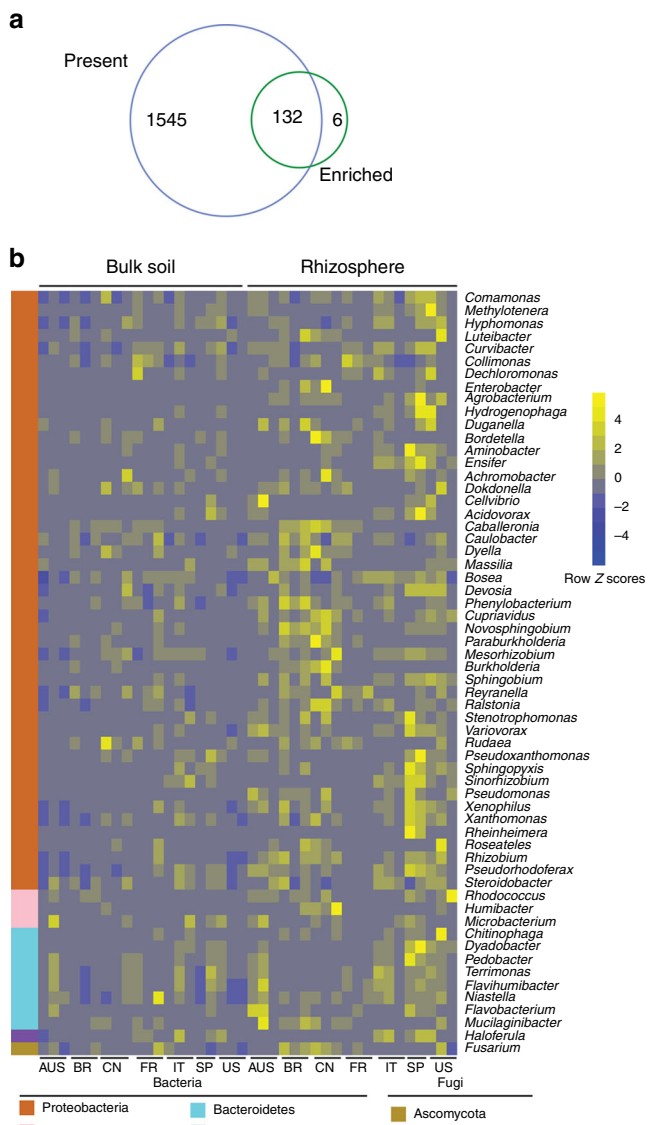

**Fig. 3** Characterization of the core citrus rhizosphere microbes. **a** Venn plot depicting the number of core rhizosphere genera based on the presence rate (>75%) in all samples and the genus enrichment from bulk soil to rhizosphere using metagenomic sequences. **b** Relative abundances of the 60 most relatively abundant core rhizosphere genera across locations and compartments based on metagenomic data. Scale, relative abundance of genus at row normalization by removing the mean (centering) and dividing by the standard deviation (scaling). The color from blue to yellow represents a relative abundance of each taxa from low to high. AUS Australia, BR Brazil, CN China, FR French Réunion island, IT Italy, SP Spain

test, Fig. 5a and Supplementary Figure 9, 10a). The antimicrobial-resistance and antibiotic synthesis genes associated-KOs were also enriched in the rhizosphere microbiome (Supplementary Figure 10b). These observations suggested that more intimate host–microbe and microbe–microbe interactions occur in the rhizosphere than in the bulk soil. Notably, the KOs involved in CRISPR-associated immunity, including Csm3, Cmr3, Csc2, Cmr6, and Csx1, were enriched in the bulk soil microbiome (Supplementary Figure 10b).

Nutrition is an important factor that shapes the rhizosphere microbiome. The rhizosphere is replete with plant-derived compounds that would be likely nutrient sources for microbes. Consistent with this, the KOs that include transporters responsible for transporting plant-derived nutrients, such as, amino acid, peptide, urea, oligosaccharide, and monosaccharide, into microbial cells, were over-represented in the core rhizosphere microbiome (corrected $P$-value < 0.05, Fisher's exact test, Supplementary Figure 11a). The enrichment of pathways involved in the degradation of aromatic compounds such as benzoate, aminobenzoate, and xylene, was also observed in the core rhizosphere microbiome (Supplementary Figure 11b). Plants would commonly release such aromatic compounds, often as defenses to plant pathogens. On the other hand, the rhizosphere-depleted KOs included those mediating carbon fixation and amino acid biosynthesis (Fig. 5b and Supplementary Figure 12). Such findings were consistent with the fact that rhizosphere microbes can acquire diverse simple carbon and nitrogen sources from root exudation[43], and thus would not need to invest in their biosynthesis. These observations were also consistent with the observation that photosynthetic microorganisms, such as Cyanobacteria were under-represented in the citrus rhizosphere microbiome. In addition, the KOs affiliated with peptidases, which are involved in protein degradation and amino acid recycling, were depleted in the rhizosphere microbiome, further suggesting that rhizosphere microbes can directly obtain amino acids from root exudates, and that amino acids represent a more important nitrogen and/or carbon source than proteins in this habitat (Supplementary Figure 13a). Transcription factors also tend to be over-represented in the rhizosphere communities (corrected $P$-value < 0.05, Fisher's exact test, Supplementary Figure 13b). Those transcription factors that were most over-represented included those in the LysR, AraC, LacI, GntR, IclR, and LuxR families of transcriptional regulators. Such regulators are typically involved in metabolism, transport, quorum sensing, motility, the stress response, and pathogenesis[44–49], processes that might be expected to be prominent in the rhizosphere.

Many of the core rhizosphere KOs might benefit plants through their involvement in multiple processes, such as nutritional acquisition, hormonal balance, environmental adaptation, and pathogenic inhibition to protect plants (Supplementary Figure 14). Specifically, some rhizosphere-enriched KOs are involved in phosphate solubilization (pqqB, appA), phosphate transport (phnCEF), nitrate/nitrite transport (nrtABC), and siderophore (iron chelating compound) synthesis. However, some KOs involved in responses to low phosphate levels (phoRPA, senX3, regX3) as well as nitrification (pmoA/amoA) were depleted in the rhizosphere microbiome (Supplementary Figure 14). The KOs involved in salicylate synthesis (ics and irp9), salicylate degradation (nagG and nagH), and acetoine/2,3-butanediol synthesis (budC) were enriched in the rhizosphere microbiome. However, some KOs involved in nitric oxide synthesis (nirK) were depleted in the rhizosphere microbiome (Supplementary Figure 14).

## Discussion

In this study, we performed a biogeographical study of the taxonomic and functional features of citrus rhizosphere microbiomes on a global scale to better determine plant driven taxa and their properties in this habitat. Previous studies of rhizosphere and soil microbiomes have been mainly based on amplicon sequencing approaches[3–6,8,10,11,14–16,50,51]. However, small-scale shotgun metagenomic studies of soil and rhizosphere microbiomes have been conducted[9,12,13,19,20] that also can provide functional information of the community in addition to community taxon composition. The decreasing cost of metagenomic sequencing has made large-scale and global studies of microbiomes in human gut[42], ocean[18], and soil[21,22] possible. Recently, Bahram and colleagues analyzed the structure and function of the

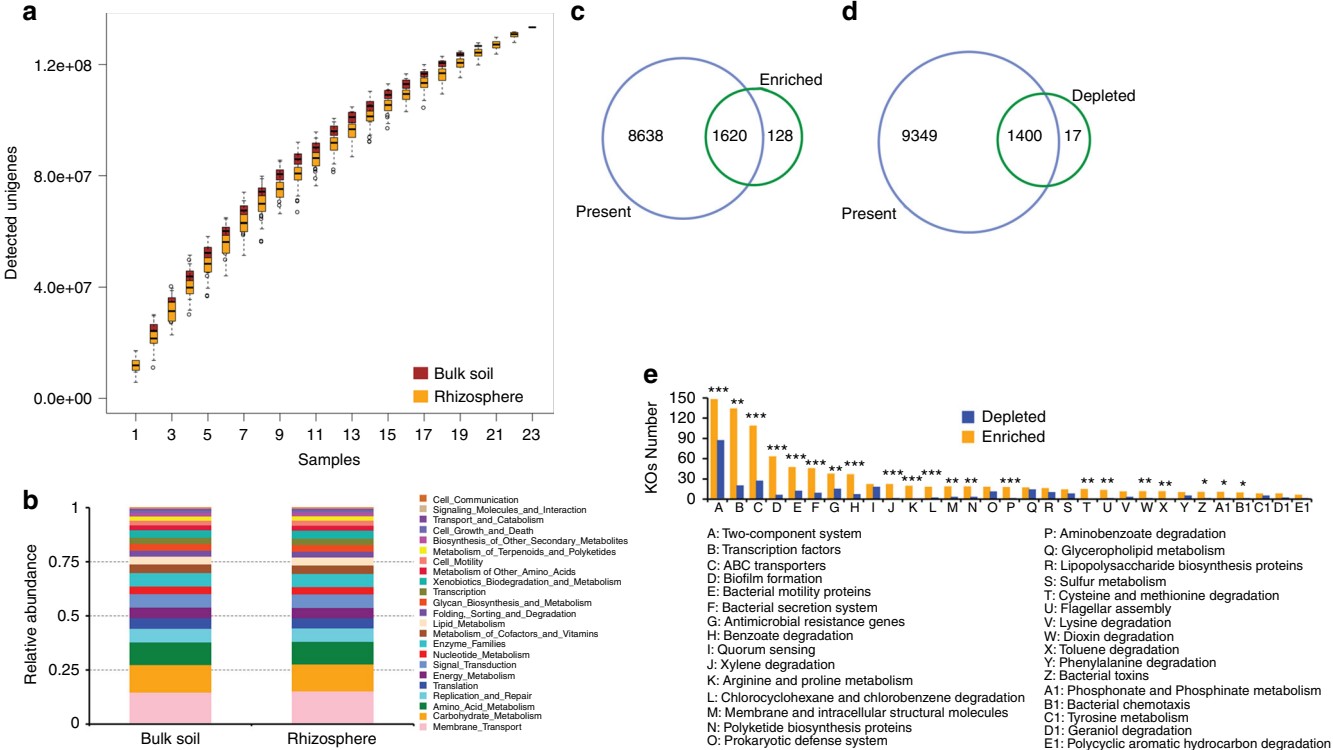

**Fig. 4** Characterization of the core function of the citrus rhizosphere microbiome. **a** Rarefaction curve of detected genes in citrus rhizosphere and bulk soil microbiomes based on 100-fold permuted sampling orders, center value represents the median of detected genes. **b** Functional KEGG level 2 pathway comparison of citrus rhizosphere and bulk soil microbiomes. **c** Venn plot depicting the number of core rhizosphere KOs based on the presence rate (>75%) in all samples and the KO enrichment from bulk soil to rhizosphere. **d** Venn plot depicting the number of universal rhizosphere-depleted KOs based on the presence rate (>75%) in all samples and the depleted KOs from bulk soil to rhizosphere. **e** The distributions of the rhizosphere core KOs in the KEGG level 3 pathways; *corrected *P*-value < 0.05; **corrected *P*-value < 0.01; ***corrected *P*-value < 0.001; Fisher's exact test and *P*-value corrected by the BH method

global topsoil microbiome using a metagenomic approach[22]. Such a study addressed the need to understand both global distribution patterns of taxa and their functional genes to better understand soil functioning[22]. In this study, we compared the structure and function of the global citrus rhizosphere with that in the associated bulk soil using both the amplicon and shotgun metagenomic sequencing approaches. There was a strong concordance between the abundant microbes detected with these two methods as in the previous studies[9]. As in earlier studies that used a metagenomic approach to profile the plant-associated microbiome[9], we found that bacteria dominated the rhizosphere and that eukaryotes accounted for a small fraction of the sequences that could be associated with known taxa. The apparently low proportional representation of eukaryotes in the rhizosphere probably results from the fact that our taxonomic classification method to identify community composition was reference-based, and the reference genomes of most eukaryotes are not available[9]. Such a conjecture is supported by the fact that more than 59% of the unigenes could not be assigned to any known taxon.

Most reports, as in the study have revealed that while there is a great diversity of bacterial communities, relatively few bacterial taxa predominate in any given soil. Delgado-Baquerizo et al. reported that only 2% of bacterial taxa account for nearly half of the soil bacteria at various sites around the globe[51]. We likewise found that only a few bacterial taxa, such as *Proteobacteria*, *Actinobacteria*, *Acidobacteria*, and *Bacteroidetes*, are abundant in both the bulk soil and citrus rhizosphere from samples taken on various continents. These abundant bacterial taxa in the citrus rhizosphere microbiome are also found to be the dominant members of the rhizosphere of other plant species[3–12,14–16]. Some

taxa from *Proteobacteria* and *Bacteroidetes* were enriched from bulk soil to rhizosphere, whereas some autotrophic microorganisms, such as *Cyanobacteria*, were depleted in the rhizosphere microbiome. The enrichment of the microbes in the rhizosphere can be attributed to their lifestyles (e.g., the fast growth and ability of copiotrophs to utilize a variety of C sources present in the rhizosphere)[21,52]. Several other reports have documented the enrichment of *Proteobacteria* and *Bacteroidetes* in the rhizospheres of different plant species[5,9,11]. Our study revealed that components of the citrus rhizosphere under field conditions were similar, despite there being present in different geographical locations, suggesting a host-driven selection for particular traits. The fact that only a few prevalent microbes are enriched in the rhizosphere from apparent soil sources, should simplify the identification of microbes as targets for future interventional studies to test their role in citrus productivity and studies in disparate locations.

The particular microbial taxa recruited to the rhizosphere from the soil microbial reservoir vary between plant species; however, a given plant genotype apparently selects a particular core microbiome[11,36]. The core microbiome of the plants probably contributes to plant growth[6] and health[13,35,53]. Thus far, the core microbiome of most plants has been defined based on taxonomic markers[3,4,6,11,16]. However, some have emphasized that more attention should be placed on identifying of microbes having common functions that are selected for in a given rhizosphere setting; such a function-based definition of the microbiome should facilitate efforts to manipulate communities for useful purposes[36,54]. Our comprehensive metagenomic sequencing of rhizosphere communities from diverse biogeographical regions

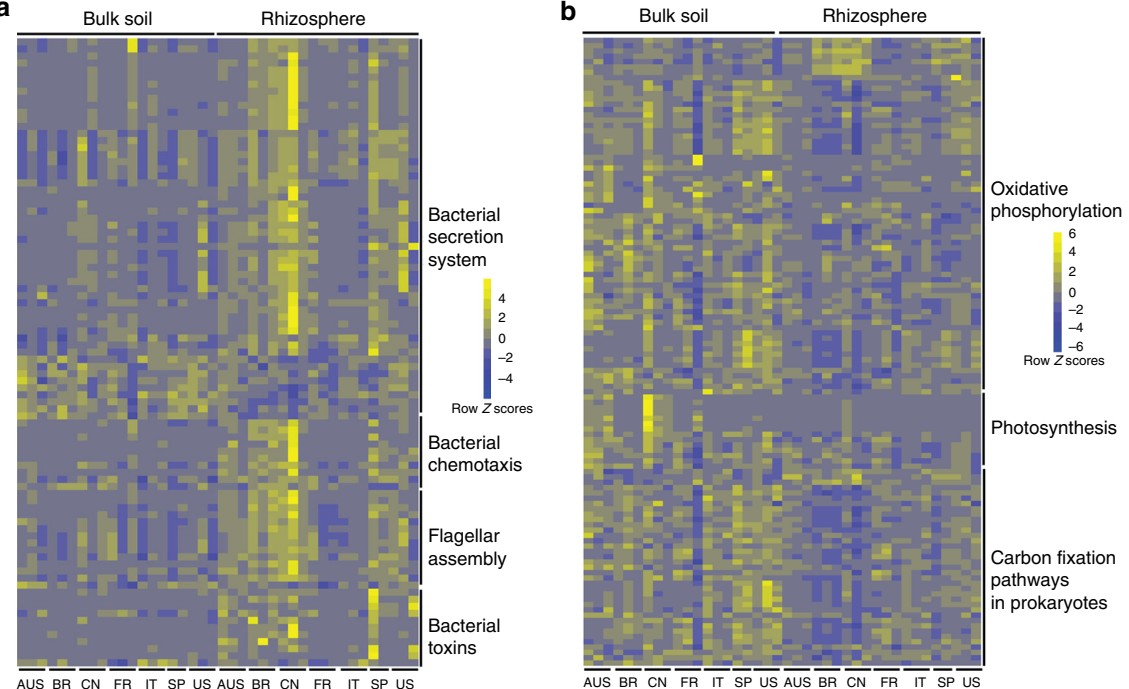

**Fig. 5** Relative abundances of core and universally depleted KOs in the rhizosphere microbiome. **a** Relative abundances of core rhizosphere KOs involved in microbe–host interaction-related pathways across locations and compartments. **b** Relative abundances of universal rhizosphere-depleted KOs involved in energy metabolism across locations and compartments. Scale, relative abundance of genus at row normalization by removing the mean (centering) and dividing by the standard deviation (scaling). The color from blue to yellow represents a relative abundance of each taxa from low to high. AUS Australia, BR Brazil, CN China, FR French Réunion island, IT Italy, SP Spain

and associated with different citrus germplasms enabled us to better define a global core of rhizosphere microbes and their functional traits. Some of the core citrus rhizosphere microbes identified here overlap with those identified in previous studies of *Arabidopsis*[3,4], millet[6], sugarcane[11], and cooloola[16], suggesting that many plant factors driving community assembly may be common between plant species. Furthermore, some of these core citrus root-associated microbes are beneficial to the plants[55]. For example, *Pseudomonas*, *Agrobacterium*, *Bradyrhizobium*, *Rhizobium*, and *Burkholderia* have all been found to inhibit plant disease in different contexts[13,35,53]. The identification of a core of rhizosphere microbes on citrus provides a useful starting point for future studies that could exploit synthetic communities to determine the interaction between microbes in their interactions with citrus itself. It will be important to determine not only the manner in which these microbial assemblages associated with the roots of citrus, but also their stability before one can associate them with, and perhaps exploit them for, stress tolerance, and disease resistance. Although we defined the core rhizosphere microbiome under the aforementioned criteria, some of these core rhizosphere microbes may be common in all soils where citrus is planted but may not be specific for citrus. Consequently, further experiments are needed to define the specific core citrus rhizosphere microbiome.

Many of the core functional traits that are over-represented in the citrus rhizosphere microbiome can be logically associated with their growth and survival in the chemically-distinct environment mediated by root exudates of citrus as well as the enhanced microbe-microbe interactions that would be expected to be present in the communities of higher cell density found in the rhizosphere. In contrast, as seen in other studies[9,12,43], functions related to carbohydrate metabolism and amino acid metabolism are under-represented in the rhizosphere core microbiome. This would suggest that the resources and

microenvironment provided by plants does not differ much between plant species. The rhizosphere enrichment of bacterial secretion systems, chemotaxis, flagella, assembly, nutrient transporters, antimicrobial resistance, and antibiotic synthesis genes indicates that the coevolution of host–microbe and microbe–microbe interactions can be logically linked to the conditions present in the rhizosphere, thus accounting for their positive selection[56]. It is therefore expected that rhizosphere enrichment of transcriptional factors would also be associated with such microbes enriched in the rhizosphere because they would be required for proper expression of adaptations to this habitat. Interestingly, some CRISPR-associated proteins were enriched in the bulk soil microbiome, indicating that microbes face more intense selection pressures from bacteriophages. Phage infection might be expected to be more prominent in rhizosphere environments due to their higher population sizes, allowing epidemics of viral infection to occur. Consistent with the identification of potential plant beneficial microbes in the citrus rhizosphere, the core functional traits of the citrus rhizosphere microbiome are likely involved in enhancing nutrient uptake by plants as well as modulating hormonal balances, thereby influencing environmental adaptation and the prevention of pathogenic infection in plants. This observation supports that core rhizosphere microbes provide benefits to plant growth and health[36].

While this study has provided a comprehensive taxonomic and functional biogeographical analysis of the citrus rhizosphere microbiome, such studies are still at a very early stage. Although the current metagenomic sequences provide some insight as to the potential functions of the rhizosphere community, assessments of interaction between microbes in plants will only be understood when we have better understanding of the expression of these traits in situ such as meta-transcriptome[12,13], meta-proteome[57], and meta-metabolome[58] data analyses and the study

of culturable members of the community[43,59,60]. In the long run, a better understanding of the plant microbiome should enable the utilization of such microbes to improve citrus health and productivity.

## Methods

**Sample collection**. To represent the biogeographical differences in rhizosphere communities, we collected samples from 23 locations in eight citrus producing countries[33] spanning all six citrus-producing continents. The samples were collected from diverse citrus varieties, soil types, and climate types and based on other soil characteristics (Fig. 1, Supplementary Table 1 and Supplementary Data 1). Representative citrus rhizosphere and corresponding bulk soil samples from these locations were collected uniformly by all participants using the following protocol. For each citrus producing country, three representative geographical locations with prominent citrus production were selected. One representative commercial grove from each location was chosen for sample collection. Four healthy citrus trees were selected from each selected grove, and rhizosphere and corresponding bulk soil samples were collected from each tree. The rhizosphere and corresponding bulk soil samples of four trees from the same grove were pooled together as one sample for each location. For each tree, the samples were collected from the 4 ordinate directions approximately 1 meter away from the trunk as shown in Supplementary Figure 1. The top 5-cm of soil was removed, and fine roots (approximately 1 mm diameter) from a depth of 5–15-cm were collected. The roots were removed from the soil with a shovel and then gently shaken to remove the soil that was not tightly attached to the roots. The roots from the four locations were pooled and washed using PBS buffer. The soil that was washed off from the roots was poured into a 50-ml Falcon tube, centrifuged and stored at 4 °C until DNA extraction on the same day; this soil was termed the rhizosphere compartment. Soil from the same 5–15-cm depth in locations without any roots was collected from multiple sites near the selected trees and stored at 4 °C until DNA extraction on the same day, and was termed bulk soil. In total, we obtained 23 rhizosphere soil samples and 20 bulk soil samples from 23 representative locations in the USA, China, Brazil, Spain, Italy, Australia, France, and South Africa, (no bulk soil samples were collected from the locations in South Africa) (Fig. 1 and Supplementary Table 1).

**DNA extraction and sequencing**. DNA was extracted from each sample using a MoBio Powersoil DNA extraction kit (MoBio Laboratories Inc. Carlsbad, CA, USA) following the manufacturer's instructions with some modifications. The DNA quality and quantity were determined by using a NanoDrop device (Thermo Scientific, Wilmington, DE) or other similar equipment and electrophoresis (0.8% agarose gel, including a 1 kb plus ladder). The DNA samples from the four trees collected from the same grove were pooled together and stored at −80 °C until use. 16S and ITS2 amplicon and metagenomic library preparation and sequencing were performed according to the manufacturer's protocol at BGI-Shenzhen, China. For the amplicon library preparation, the amplification of 16S and ITS2 DNA fragments was performed using the common amplified primers and methods of prokaryotic 16S rDNA V4 region (515F and 806R)[61] and fungal ITS2[62]. For metagenomic library preparation, the metagenomic DNA was sonicated to the 350-bp size range. DNA fragments were then end repaired, 3′-adenylated and amplified using Illumina sequencing adapter-specific primers. After quality control, quantification and normalization of the DNA libraries, 150- and 250-bp paired-end reads were generated from the Illumina HiSeq4000 and MiSeq platform according to the manufacturer's instructions with modifications for the metagenome and amplicon (16S and ITS) analyses, respectively. More than 30 Gb of clean data and 30,000 clean reads were generated for each metagenomic and amplicon sample, respectively (Supplementary Table 2–4).

**Amplicon data analysis**. Microbial community composition was determined by sequencing 16S rRNA gene and ITS2 amplicons from DNA samples from the citrus rhizosphere and corresponding bulk soil. The high-quality paired-end reads of the 16S V4 region and ITS2 were merged using FLASH software with the default setting[63]. The OTUs were obtained using UPARSE pipeline[64] based on the merged sequences. To obtain the taxonomic information of the OTUs, representative sequences of each OTU were generated and aligned against the SILVA[65] and UNITE[66] databases using the RDP classifier[67] for 16S and ITS2, respectively. The OTUs and merged sequences, which were defined as unknown, chloroplast, mitochondria or plants, were removed. The relative abundance tables for taxa (OTU, genus, and phylum) were generated based on the read count for each taxon across samples by using the total-sum scaling (TSS) method[68]. Within-sample diversity was calculated for each sample using the Shannon index based on the normalized OTU abundance table using the rarefied method[68]. The significant differences in alpha diversity across compartments were determined using two-way ANOVA and one-sided t-tests[9]. The taxonomic dissimilarity analysis between samples was performed based on the PCoA method with unweighted UniFrac distances (beta diversity)[69]. The VPA analysis with two-way PERMANOVA was carried out based on the OTU relative abundance table using VEGAN packages in R software[9].

**Metagenomic data analysis**. The raw reads from metagenome sequencing were used to generate clean reads by removing adaptor sequences, trimming and removing low-quality reads (reads with N bases and a minimum quality threshold of 20) at BGI-Shenzhen, China. The clean reads were further trimmed using Sickle software[70], and trimmed reads shorter than 80 bp were discarded. The trimmed reads were mapped to the sweet orange[27], *Citrus clementina*[28] and Swingle citrumelo[71] genomes using Bowtie2 software[72] to identify and remove the citrus host-originated reads. The 23 rhizosphere and 20 soil samples were separated into 9 groups (Supplementary Figure 2) based on their microbial community similarity calculated by Mash[73], and the pooled metagenomic reads from each group were de novo assembled using Megahit[74] ver. 1.0.3 with the meta-large preset parameter (Supplementary Table 5). The final assembly comprised 223,971,928 contigs, with a total length of 133.4 Gb. The metagenes were predicted using Prodigal[75]. Using CD-HIT-est with the identity cutoff of 95%[76], 183 million nonredundant metagenes (unigenes) were obtained. To generate the taxonomic information of the unigenes, the protein sequences were aligned against the NCBI microbial NR database, which included bacteria, archaea, fungi, virus, protozoa, algae, and plants, using DIAMOND[77] software with an *E* value cutoff of 1e−5. Based on the MEGAN LCA algorithm[78], the taxonomic annotation of the unigenes was assigned. To obtain functional information for the unigenes, the protein sequences were blasted against the KO database using DIAMOND software. To generate the taxonomic and functional abundance profiles, the reads from 23 rhizosphere and 20 soil samples were aligned to the unigenes using SOAP2[79] with the default setting. The generated alignments were parsed, and the read count abundance was generated.

**Comparison analysis across compartments**. Based on the abundance profiles, the features (genera, phyla, and KOs) with significantly differential abundances across compartments were determined using a statistical method, such as DESeq2[68] with a negative binomial generalized linear model. The read count matrix for DESeq2 testing was normalized using the DESeqVS method[68]. For the detection of rhizosphere-enriched genera (abundance significantly higher than that in bulk soil), rhizosphere-depleted genera (abundance significantly lower than that in bulk soil) and KO in the metagenomic data, paired DESeq2 comparison analysis was performed based on the read count matrix of the genera and KOs across the bulk soil and rhizosphere samples ($n = 20$ for each group). To determine rhizosphere-enriched and rhizosphere-depleted genera detected in the 16S ($n = 20$ for each group) and ITS2 ($n = 13$ for each group) data, we also used the paired DESeq2 comparison analysis method. *P*-values for multiple testing were corrected using the BH method in DESeq2. All items with corrected *P*-values < 0.05 were considered significant. Furthermore, we defined core citrus rhizosphere microbial genera and core citrus rhizosphere KOs based on the following: the genus or KOs that were present in more than 75% of the samples across-the-globe for each group and that were statistically enriched in the rhizosphere samples compared with the corresponding bulk soil samples[6]. The relative abundances of rhizosphere-enriched or rhizosphere-depleted taxa and functional traits are shown using the Pheatmap package in R software. To demonstrate a clear rhizosphere-enriched/depleted pattern, the relative abundance of each taxon or functional trait was normalized by removing the mean (centering) and dividing by the standard deviation (scaling).

**Code availability**. Bioinformatic code is freely accessible through our website (https://db.cngb.org/icrm).

## Data availability
The raw sequencing reads were deposited in the NCBI Bioproject database under the accession number PRJNA362455. The nonredundant reference catalog of the citrus rhizosphere and bulk soil microbiome is freely accessible through our website (https://db.cngb.org/icrm). Other data supporting the findings of the study are available in this article and its Supplementary Information files, or from the corresponding authors upon request.

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

## Acknowledgements

This work has been supported by Florida Citrus Initiative. We thank Drs. Steven E. Lindow and Frank White for their critical reading and help with the manuscript. We thank Dr. Wenbo Ma and Kelley Clark for their help with sampling.

## Author contributions

N.W. conceived and supervised the project. J.X., Y.Z. and N.W. designed the experiment. Y.Z., N.W., N.R., P.T., J.T., H.D.C., J.C., X.D., V.A., Z.L., B.Z., C.R., N.C., V.C., G.P., C.V., A.M.A. and L.L. collected samples and extracted DNA. J.X., Y.Z., P.Z., Y.W., X.L., G.F. and F.Y. analyzed the data. J.X., Y.Z., P.Z. and F.Y. constructed the database. J.X., Y.Z., J.T. and N.W. wrote the manuscript. P.T., Y.W., X.L., G.F., J.C., C.R., G.P., C.V., X.X., J.W. and H.Y. revised the manuscript. All authors read and approved the final manuscript.

## Additional information

**Competing interests:** The authors declare no competing interests.

