## [Peer Review File · Nature Communications]

Reviewers' comments:

Reviewer #1 (Remarks to the Author):

The manuscript by Xu et al. describes the structure and function of the citrus root and rhizosphere microbiomes in comparison with the corresponding soil microbiomes. The work is highly representative with a global sampling effort, is timely and the authors utilized appropriate methods. The manuscript covers an impressive amount of work coupled with a deep functional analysis. I appreciate that the citrus rhizosphere microbiome was compared to ocean and human gut biomes and was also benchmarked with other plant species. Such efforts are needed to widen the inference space and to make a manuscript attractive for a broad readership. This work has certainly a great potential to be interesting for a general audience, however, I recognize a substantial amount of major and minor concerns to be clarified as well as issues for improvement.

Major comments:

Manuscript structure: I am not convinced about the manuscript organization. Intuitively and following the manuscript title with '...structure and function...', I would first report taxonomic composition (amplicon data and metagenomes) and then the functional and genomic aspects (metagenomes and MAGs). Also, the comparison of CRAM with other plant root microbiomes (after MAGs) appears off a logic manuscript flow. Similarly: the (sub-)Figure order does often not match the manuscript text. For instance, Fig. 3 contains the taxonomic split up by location (subfigures B – D), which is reported in the text after Figures 4 and 5 are described. Or the phylum level statistics (referring to Fig. 3A) is reported at L213 after describing the diversity analysis and Figure 4. Another example: Fig. 6F is followed by 6E.

Metagenome comparison: I feel uncomfortable with the citrus root-associated soil microbiome (CRASM) merging the two distinct biomes of the citrus rhizosphere and its corresponding soils. I would keep and treat the rhizosphere and soil microbiomes separately. I expect the removal of citrus host reads (L118) to be different for the two sample types. I consider the identification of core functional traits for the citrus rhizosphere (compared to soil) appears the most important finding and advance of the manuscript (Fig. 6). To further highlight this, a side-by-side graph of soil vs. rhizosphere would be more appropriate than splitting the data by location (Figs. 6A&B), which is not discussed anyway in the results section.

Terminology: The manuscript uses unnecessary complicated and sometimes confounding terminology. It is not immediately clear to what the authors refer to (e.g., does the expression 'root-associated soil microbiome' cover the 'rhizosphere', the 'root plus the rhizosphere', 'rhizosphere and soil'...?). I suggest to refer directly to the investigated sample types and simply term them as 'soil', 'rhizosphere' and 'root'. By the way, 'layer' is an uncommon term, 'compartment' is also widespread. I also felt that defining the many and similar abbreviations 'CRASM', 'CRAM' and 'CCRAM' does not make the manuscript easy accessible to readers. I do not see the point for introducing these abbreviations, especially also because the authors do not employ them in the discussion of their work. A general remark regarding abbreviations: define all abbreviations (e.g., PCoA, ITS, ARGs, ...) but only define abbreviations, which you often use in the MS (e.g., VPA is defined but not used).

Diversity analysis: I have a number of comments to the alpha and beta diversity analysis investigating the effects of sample type and location (L199...; Fig. 4A and B). The separate panels for sample type and location preclude to see possible interaction effects. I suggest to combine the two figures in one, were the three compartments are shown for each site. The logic order for sample type is 1) soil, 2) rhizosphere and 3) root. I do not see a decrease in Shannon diversity from soil to the rhizosphere (L203). The support for the claim in L204-207 is not easily documented as the distinction between Fig. 4B and S8 panel figures has to be made by reading the captions. Is Fig. 4B a mix of the sample types? What is about the fungal data? The 3

dimensions of the ordination graphs cannot be interpreted on 2D. It only makes sense to report PCo axes 1 and 2 in a printed graph and if helpful, axes 2 and 3 in a second 2D graph. I am confused why the ordination graphs change between Fig. 4C and 4D. The samples points should stay in place in an unconstrained ordination and simply the color coding should change whether sample types or locations are highlighted. Please clarify if not eventually constrained ordination was applied. Fig. S9 is then redundant.

Kruskal-Wallis test: The authors utilize the Kruskal-Wallis test to identify location-specific taxa (L228...). This test is revealing, if there is any difference between the tested locations but does not identify which location(s) is/are different from each other. For the latter purpose, a dedicated non-parametric post-hoc test is required. Inspecting the Supplementary Data 3, none of the phyla has a significant adjusted P-value for soil and rhizosphere samples – hence the conclusion is that there are no differences between the locations!

Balanced design: Without going to the method details, the reader gets the impression that the three sample types were collected at all 28 sites (L111). However, especially for soil (20 samples), the experimental design appears unbalanced. I could not square how Table S1 (23 locations) compares to the mentioned 28 sites in Fig. 1. Please clarify where samples are missing.

Sampling depth: The CRASM has a good global sample coverage while the representativeness for the ocean and human gut metagenomes remains unclear. Can you place the 4x and 8x differences in gene coverage (L124) in scale to the ocean and human gut examples? How representative are the ocean and human gut metagenomes (worldwide and diverse sampling?). Along the same lines, I was wondering if the comparison for differences in gene coverage takes possible differences in sequencing depth into account (the deeper you sequence, the more new genes you will find).

Network analysis: People tend to over-interpret co-occurrence network graphs. The correlation analysis recapitulates simply the abundance behavior of microbes between two sample types: e.g., abundant taxa in soil positively correlate with each other and abundant rhizosphere taxa co-occur with each too, while negative correlation are mostly found between soil and rhizosphere-specific groups. This is not surprising, but this lies in the nature of the analysis with the correlation pattern reflecting the condition of the comparison (here: rhizosphere vs. soil). L297: explain, why the opposite abundance behavior of the mentioned taxa is intriguing. L302 and 305: refrain from interpreting 'interaction' or 'communication' between microbes as such correlation based analyses simply reveal co-occurrence patterns. In a way, co-occurrence networks can be seen as maps of abundance behavior for a certain condition (soil vs. rhizosphere).

Graphic help: In general, it would be a service to the reader if the type of compartment (soil, rhizosphere or root) and type of data (amplicon, metagenome) is indicated in all graphs and there is no need to retrieve this information from the caption

Code: all bioinformatic code must be made publically available!

Minor comments and suggestions:

- The manuscript contains numerous language 'bumps' and typos. E.g. in L42, "The plant root-associated plays..." or "Plant root-associated microbiomes play_...". Similar in Lines 60, 57, 68, 128...)
- L49: You refer to the human gut, no?
- L50: Avoid unexplained methodological information (identity and coverage).
- L54: Rephrase: "...the amplicon of prokaryotic and fungal ribosomal operon marker genes."
- L62: To which soil properties do the authors refer to?
- L68: Avoid dual mentioning.
- L77: The authors report that the PCR-based taxonomic profiling is similar with the unbiased shotgun metagenomic taxonomy. I was wondering, if it is necessary here to raise the possible

issue of PCR biases of amplicon approaches? Add references or examples for the 'PCR bias' and 'niche-specified' claims.

- Table S2: typo in the header ("sickle treated sington"), which samples are rhizosphere and soils?
- Fig. 1C: I greatly appreciate the metagenome comparison of the citrus rhizosphere and corresponding soil to ocean and human guts. Why is human gut missing in this figure?
- Fig. 1E: I do not see the need for this figure as this message (L127) is part of Fig. 1C and because annotation differences between locations were not statistically investigated or reported.
- Fig. 2A: Use one color with different hues for the various classes of Proteobacteria. This will help the reader to appreciate the Proteobacteria claim in L139 more easily. I could not find a table that supports the statistic analysis. I suggest to add an asterisks to the taxon name in the legend for the taxa that are significantly different between the three biomes. Update the Fig. 2C in a similar manner with the statistic information.
- Fig. 2B: I would use more intuitive colors: blue for the ocean and green or yellow for citrus.
- Fig. 2E/ Fig. S4: Ordination graphs would be more straight-forward to read.
- L149-151: add reference to Fig. 2D.
- L181: While the authors mention the average sampling intensity for the metagenomes, this is not done here for the amplicon approach.
- L182: Explain how the metagenomic data was used to assist with the taxonomic analysis.
- L249: Acidobacteria were markedly lower in abundance compared to Proteobacteria and Actinobacteria (Fig. 3A) and I was wondering if the authors can explain, why most of the MAGs belonged to Acidobacteria? Differences in genome size, less complex assembly, fewer repeats, ...?
- L252: I am surprised that 42 (10%!) of the MAGs could not be assigned even at highest taxonomic phylum rank. Can you exclude technical artifacts?
- L276: what groups? The sample types?
- L289: Fungal data not inspected?
- L290: I would be very careful with the 'beneficial' claim. Pseudomonads as well as Agrobacteria comprise well-known plant pathogens. The metagenome further contains Ralstonia or Xanthomonas!
- Fig. 5A/B: What is the taxonomic/sequences-based overlap/agreement between 16S and metagenome data?
- Fig. 5C/D: Explain the scale (negative values) for the relative abundance. Too many colors to be discriminated. Label the locations with the previously used abbreviations (Aus, Br,...), I would also label the sample types without a color code. I guess the left-hand color bar should refer to phylum level taxonomy? Legend is missing. The same comments apply to Figures 6E and F.
- L292: I am confused how the MAGs are linked with the CCRAM?
- L363: With regard to microbe-microbe interaction, any indication of type-6-secretion system increase in the rhizosphere?
- L383-387: This claim is strongly linked to sampling intensity. The authors have 16S data with a sampling depth of 20ish thousand reads per sample, which is rather low for soil profiling.
- L398: Rephrase to "Citrus is planted worldwide and subjected to nutrient limitations ..."
- L407: It remains unclear how the citrus microbiome data could be deployed for synthetic microbiome engineering. Examples or ideas are needed. In particular, how should network analysis help to predict inoculation success of microbes to agro-ecosystems?
- L525: Explain how the reads were "cleaned" at BGI?
- L566: 23 rhizosphere and 20 soils samples

Reviewer #2 (Remarks to the Author):

The manuscript “The structure and function of the global citrus root-associated microbiome” by Xu, et. al. attempts to characterize the taxonomic and functional components of the citrus rhizosphere microbiome through the use of metagenomic analysis. This manuscript represents a considerable portion of work and is based on a rather a large collection of citrus rhizosphere samples from throughout the world. However, there are several issues concerning the writing and interpretation of the results could use extensive revision.

My concerns can be listed as following:

- 1.) Focusing on microbiome functional over-lap between the plant rhizosphere with ocean and human gut micro-environments is inappropriate.
- 2.) A lack of description of the germplasm and soil characteristics in samples used in this study.
- 3.) A lack of discussion regarding the disadvantages and assumptions that are inherent in metagenomic studies of the microbiome

The author’s make dubious claim that the human gut is functionally analogous to the rhizosphere and support the claim with a single review article. However, there are multiple published empirical studies that support that fact that very few microbial taxa overlap between these environments (see Thompson, Nature, 2017 for a recent example) suggesting that there should be very little functional overlap in the metagenomes of these environments. Additionally, the human gut a relatively consistent environment with acid pH and temperature and is far more resilient upon perturbation with diet (David et. al, Nature, 2014) where as the authors freely admit that the rhizosphere is far more variable and is more strongly influenced by factors such as “geographic locations, salinity, temperature, oxygen, nutrients, pH, day length, and diseases^{20,23-30}” which influences microbial community composition and function. The authors also make no justification why they compared rhizosphere metagenomic data with oceans, which is also widely accepted as being drastically different in microbial composition. Thus, the mere fact that metagenomic analysis of citrus rhizosphere’s differs from human and ocean environments is neither surprising nor novel. The manuscript would be better served by focusing on the advantages and disadvantages of metagenomic analysis for understanding community function and that this paper focuses on comparing a plant rhizosphere community (which has both free-living and host-associated components) with two well-characterized metagenomes (one free-living, i.e. ocean dataset, and one host-associated, i.e. human gut).

The authors also do not adequately describe the level of natural variation in the citrus host used for this study within the manuscript and have hidden this information in the supplemental tables. Presumably, these are all agricultural fields that are using a narrow subset of domesticated citrus varieties grown on cultivated plots. Citrus is well-known to produce a wide of terpenoids that varies widely both qualitatively and quantitatively among both domesticated and natural citrus varieties. Therefore, it is unclear if some of the rhizosphere enrichment observed in the analysis is common to a wide variety of citrus genotypes or a function of only a single citrus variety that is widely cultivated throughout the world. Additionally, it is unclear if the geographic differences

highlighted in the results are due to abiotic soil conditions at that site or plant varietal differences for citrus that is commonly cultivated within that region of the world. This lack of detail also obfuscates the general utility and extensibility of this analysis for both citrus and other crop species.

Lastly, both the introduction and discussion lack any consideration of the potential limitations of the method in addition to advantages. Since metagenomic analysis relies on the enrichment of DNA in a particular environment, results often strongly correlate with taxonomic enrichment regardless of the transcriptional/translational state of the gene in question. For example, cephalosporin-biosynthesis identified in this manuscript is common among proteo- and actinobacteria and thus should be enriched in the rhizosphere environment. However, this genomic enrichment does not mean that these pathways are expressed and utilized under within this micro-environment and also cannot be used to determine if these are used in either the biosynthesis or degradation of cephalosporin-like secondary metabolites. This criticism of the technique applies to interpretation of all results from genomic enrichment of biochemical pathways at the DNA level. The authors need to either: 1.) control for taxonomic enrichment in this analysis, or 2.) address the limitations of the method in either the introduction and/or the discussion.

Detailed Comments:

Abstract:

42: **may* play roles Not under all conditions and are likely small factors relative to larger trends.

43-44: "... are largely unknown." The taxonomic contents of plant rhizospheres are fairly well characterized, as cited by the authors.

47-52: Why are the authors comparing their metagenome analysis to human and oceans? I'm sure that there are other previous soil metagenomic papers to compare their results to (e.g. Fierer 2012 PNAS). The scope of comparisons is inappropriately wide. See above.

55: Is the core root-associated taxa specific to citrus or have the authors just re-identified rhizosphere specific taxa.

Introduction:

The whole first paragraph could be removed without much impact on the remainder of the story. It jumps around in focus and does not present the following work well.

59: The animal gut is not widely considered to be analogous to the soil rhizosphere. See above.

63-67: This is more of a conclusion and potential broader impacts statement.

75-79: The authors comment on the limitations of 16S DNA profiling, but fail to mention the limitations in metagenomic analyses, such as KO-enrichment closely tracks with microbial phylogeny and should not be extrapolated to believe that enriched genetic signatures at the DNA level are functional in that environment. See above.

87-93: This should be the beginning of the introduction. The focus of the study is on citrus and NOT on plants in general.

Methods

The sample numbers don't always add up and/or it is unclear how and when the authors attempted to pool. For example:

448: The authors list 71 samples split among roots, rhizosphere, and bulk samples, but previously stated that they sampled 4 trees from 28 sites across the globe, adds up to 112 samples not including the bulk soil. Please clarify.

528-529: It is unclear what samples are within the 43 samples listed for metagenomic analysis. Presumably, it refers to the rhizosphere and bulk samples. Please make explicit.

Also:

445-447: Please expand on the bulk soil site identification and collection.

452-489: Place in supplement and only restrict the deviations from the manufacturer's protocol for the main text.

496-498: The authors MUST expand on the how the DNA libraries for the metagenomic sequencing was made. A part of a sentence is wholly inadequate.

508: I would suggest the authors use a denoising focused pipeline, such as uNoise3 or DADA2, to remove sequencing errors from their 16S analysis.

510: Please cite the reference databases used.

513: Please cite the TSS normalization method used.

514: the Shannon index should only be calculated on rarefied data since more species will be identified depending on sequencing depth and normalization can skew presence/absence data used to calculate alpha-diversity.

542, 566-567, 576-577: There doesn't seem to be any normalization method for the metagenome analysis. Thus, enrichment analyses could very well be the function of sequencing depth artifacts. In addition, I don't see how the authors controlled for multiple testing? The results mention FDR corrections, but should be described here.

582: I would recommend running the permutation 1,000x. Thresholds can be arbitrarily low with inadequate replication in permutation estimation.

595-596: A threshold of $r \geq 0.7$ seems extremely low and only translates to a nominal R^2 of 0.49. This likely results in excessively connected network for the given data.

Results

Figures are often unnecessarily complex or not properly referred to in the text.

Fig. 1C: out of order relative to the text and lacks an analysis of the human gut. Also, since the rarefaction curve suggests that the metagenome is under-sampled, the proportion of reads assigned to viruses and eukaryotes/protists for the citrus rhizosphere is not likely representative. Also, the authors should define how they are describing the last universal common ancestor (LUCA) in this context. Move to supplement or remove completely.

Fig. 1D: By including the combined 'All' unigene category in the graph obfuscates the fact that neither the rhizosphere nor the bulk soil are close to saturating the potential unigenes for the microbial community. While the authors do state this, the graph is wholly misleading as is.

Fig. 1E: would be better described as a table rather than a graph. Move to supplement.

127: needs to state the variance in unknown genes across the sites sampled.

133-137: Analysis out of place. Also these results are from the metagenomic analysis and not the 16S profiling. The remainder of the paragraph is not clear if the taxonomic profiling is coming from the 16S sequencing or from a phylogenetic analysis of the metagenomic assembled genomes.

Fig 2A&B: please label if these are from 16S profiling and/or the metagenomic analysis.

144: This is not necessarily noteworthy as this has been shown consistently. Most recently by Thompson et. al. Nature (2017)

149-174: Please also compare these results with other soil metagenomic analyses, such as Fierer et. al. PNAS 2012. The bulk soil results should compare directly.

151: "The KOs..." the CRASM KOs?

154-155: Speculative and obvious.

156-159: Difficult to follow and a run-on sentence. Split up.

Fig 2E: move to supplement

170-174: This is a ludicrous claim that the soil is the source of inoculum for oceans and the human gut and doesn't follow any known ecology or evolutionary theory. Especially for humans guts with individuals from developed nations where there is a long production line of food washing/processing/inspection/storage between the soil and human consumption. More likely,

these are highly generalist taxa that are highly plastic in their ecological niche and are extremely ubiquitous in all microenvironments, including the human gut and oceans.

175-179: It is difficult to parse the difference between the goals of the citrus root associated soil microbiome (CRASM) and the citrus root associated microbiome (CRAM). Please use a different description and acronym.

184: “amplicon”? do you mean 16S profiling?

186-187: Please use the coefficient of correlation (r^2) for Spearman correlations instead of the coefficient of determination (r). Use of r is misleading and gives the impression of a stronger relationship than actually exists.

191-216: Figure order is confusing. The authors are asking the reader to jump around the figures often and is making it difficult to follow the logic. I would suggest restructuring the figures according to logical flow of the claims rather than by similar analysis.

204: Given the complex influence of microcosm and location on diversity, Kruskal-Wallis is the wrong test. I would suggest at least a two-factor linear model with passion link function for Shannon Index or equivalent non-parametric test. This should more closely correspond to the VPA results. Also, this should be done on rarefied data, not TSS transformed data. See comments on methods.

213-239: Same issue as above. The authors need a more flexible statistical framework. Also, they should incorporate a multiple testing correction to reduce their false-positive rate.

228-229: How do these site-specific signatures correspond with the soil characteristics from each site? For instance, Brazilian soils are often contaminated with high levels of aluminum, which can limit calcium availability in the soil community. This is especially important as many studies have shown that soil abiotic conditions are more important for microbiome taxonomic composition.

242-246: What is not clear to me is if the MAGs were constructed within site or across the entire set. If the later, what assurances do we have that the in silico constructed MAGs are not just a bioinformatic artifact? To address this, the authors should state what percentage of the reads for a MAG are from the largest contributing site (cis- reads) vs reads that came from other sites (trans-reads).

266: “...higher *taxonomic* resolution...”

262-273: This should be moved up in the results to justify that these results are similar to what has been seen for other model systems. The authors should also mention if there are any taxa that appear to be qualitatively specific to the field citrus rhizosphere. If not, please state explicitly.

274: Please don't use this acronym (CCRAM). It's just confusing with CRAM and CRASM and it makes it hard for the reader to keep follow what community is under discussion. Just leave it as the core rhizosphere community.

Fig 5A and B: Please label the Venn diagram with compartment/method.

Fig 5C and D: Please label the top and bottom of the heat plot with names instead of colors as this is difficult to follow and is confusing for a color-blind audience.

285-290: Are these core taxa common members of core communities for other plant species or are these specific to citrus?

291: Is this relative to the 16S data or the metagenomic data?

292: Again, please use r^2 and not r for correlations.

294: Network robustness is not a function of the significance of the edges. It is the ability of the network to maintain its topology, structure, and/or output based on perturbation or removal of individual nodes and/or edges. Please remove as this statement is not supported by the data.

302: "...interact..." change to "associate"

305-309: The authors make extremely broad assumptions to conclude that core microbes "... tend to communicate with other core members, probably through their metabolic handoffs. ..." from a simple correlational relationship. They could also just respond similarly to environmental cues, e.g. increasing root exudate concentrations, which would also result in a significant correlation.

Fig 5E: Please label the left and right side of the dotted line. Also, use a different color for negative correlations. Also, the authors did not filter the network for the same correlations, i.e. the correlation of A to B is the same as B to A. This is most obvious with the singleton correlations, such as *Caldisericum* with *Planktothricoides* and *Pseudoduganella* with *Duganella*. Please filter.

Fig 6A and B: uninformative, move to supplement.

Fig 6C and D: Again, please label the Venn diagram. Present in what? Enriched in what?

Fig 6E and F: Please label the sides and do not use the color system. Also, please split up the legend because it is confusing to relate compartment with site.

Fig 6G: Again, please filter the network edges for forward and reverse relationships and use a different color for negative correlations. Also, please label the subnetworks that are being highlighted.

325-327: Please expand on why metabolic pathways, such as carbohydrate and amino acid and energy metabolism would be depleted in a carbohydrate/amino acid/energy-rich environment like the rhizosphere. This is not an initially intuitive result for me.

332: How are the authors defining "positive selection" in this case? The analysis doesn't seem to conform to analyses of natural selection from population genetics that I am familiar with.

334: Please define ARGs.

335-338: It is possible that this enrichment is due to enhanced antibiotics used by these two countries but this could also be a function of anti-biotic production from other microbes or fungi

within the soils from those environments. If the authors have meta-data or a study that supports the claim that Brazil and China add exogenous antibiotics to combat common citrus disease, please cite.

Discussion

340: change “recruit” to “enrich”. Also, “huge” is a bit of an overstatement considering only 50+ genera make up the core of the community.

340-355: The authors readily list the limitations in other studies and approaches but fail to acknowledge the limitations of their own. See above comments.

356-359: The authors only seem to recognize that metagenome sequencing has occurred in mammals and oceans while ignoring previous work that has been done in other soil systems. If they believe that this study is a better, or more in-depth, metagenomic analysis, please state. The contrasting of the rhizosphere with these environments is a bit of a straw-man argument as they are enriching for differences while ignoring previous work that has been done in similar soil environments. Please make that comparison rather than these disparate studies that appear to have little to no relationship to the topic under discussion.

356-379: This paragraph reads as more results rather than shaping and contextualizing the results into a cohesive narrative with a broader perspective to literature. Consider moving portions to the results.

383-387: This conclusion is misleading as the depth of sequencing of DNA libraries for the metagenomic analysis was considerably deeper than the 16S sequencing. As shown in the rarefaction curves, both methods are still under sampling the extreme diversity present in these samples.

393-397: Personally, I question how many of these are “true” novel genomes that exist in nature and how many of these are simply artifacts of bioinformatic genome construction. Additionally, the authors should address how intra-specific and intra-generic genetic variation can affect the bioinformatic construction of hard to culture microbes. Just because a computer can generate it, doesn’t mean it is real.

420: the data “suggest”, they do not “indicate”.

422: How do these data “...underline the common mechanism used by plants to recruit microbes...”? The study only used a single plant genus and does a poor job comparing to the rhizosphere studies in other plants.

424-417: Do not bring the topic back to the larger comparisons of human and ocean microbiomes that they set out to explain in the first place. This conclusion is far too narrow given the attempted scope of the study.

Response to reviewers' comments

Reviewer #1 (Remarks to the Author):

The manuscript by Xu et al. describes the structure and function of the citrus root and rhizosphere microbiomes in comparison with the corresponding soil microbiomes. The work is highly representative with a global sampling effort, is timely and the authors utilized appropriate methods. The manuscript covers an impressive amount of work coupled with a deep functional analysis. I appreciate that the citrus rhizosphere microbiome was compared to ocean and human gut biomes and was also benchmarked with other plant species. Such efforts are needed to widen the inference space and to make a manuscript attractive for a broad readership. This work has certainly a great potential to be interesting for a general audience, however, I recognize a substantial amount of major and minor concerns to be clarified as well as issues for improvement.

Answer: Thanks for your encouragements. We revised the manuscript carefully following your comments.

Major comments:

Manuscript structure: I am not convinced about the manuscript organization. Intuitively and following the manuscript title with ‘...structure and function...’, I would first report taxonomic composition (amplicon data and metagenomes) and then the functional and genomic aspects (metagenomes and MAGs). Also, the comparison of CRAM with other plant root microbiomes (after MAGs) appears off a logic manuscript flow. Similarly: the (sub-)Figure order does often not match the manuscript text. For instance, Fig. 3 contains the taxonomic split up by location (subfigures B – D), which is reported in the text after Figures 4 and 5 are described. Or the phylum level statistics (referring to Fig. 3A) is reported at L213 after describing the diversity analysis and Figure 4. Another example: Fig. 6F is followed by 6E.

Answer: Thanks for your valuable suggestions. We re-organized the manuscript as you suggested. In the current version, we firstly showed the results of “Global citrus-associated bulk soil and rhizosphere microbiome gene catalog construction”, then presented the “Taxonomic content in global citrus rhizosphere and root microbiomes”, defined the “Core taxa of the citrus rhizosphere and root microbiomes”; next, we reported the “Comparison of rhizosphere and root microbiomes between citrus and other plants”, as well as the “Taxonomic content comparison among microbiomes of citrus-associated bulk soil, rhizosphere and other ecosystems, including non-citrus associated soil, human gut and ocean samples”. Finally, we performed the “Functional composition of the citrus-associated bulk soil, rhizosphere microbiomes and comparison with that from other ecosystems” and also defined the “Core functional traits in global citrus rhizosphere microbiome”.

As other reviewers and previous studies (Thompson et al., 2017, Nature) suggested that there should be very little functional overlap in the metagenomes of citrus-associated bulk soil, rhizosphere, ocean and human gut microbiomes, we removed the claims and discussion for microbiome taxonomic and functional overlaps between the plant rhizosphere with ocean and

human gut micro-environments since it is beyond the scope for the current manuscript. We focused on the microbiome differences between citrus rhizosphere and human gut, and between citrus-associated bulk soil and ocean. In addition, we did the comparison among the microbiomes from different soil systems, including citrus-associated bulk soil, desert and non-desert soil.

The purpose of MAG analysis was to provide cues for the bulk soil to rhizosphere enrichment process of the citrus root-associated microbiome at single genome level, and the results were consistent with those from the community-based analyses. However, we went through the manuscript carefully based on the all reviewer's suggestions, and agreed that the MAG section was out of logic, did not provide enough novel insights and information for this manuscript, and we have removed the MAG analysis in the current manuscript. In addition, we found that these MAGs represented a very small part of our data. For example, the total sequence length of all MAGs, number of predicted genes, average percentage of mapped reads were only 1.3 Gb (10 % of total sequence length for metagenomic contigs), 1.42 millions (0.6 % of 230 million metagenes) and 2.6%, respectively. Furthermore, the abundant *Proteobacteria* and *Actinobacteria* included too many similar and high abundant genomes. Based on the current methods using the tetra-nucleotide frequencies, abundance and GC contents, it's hard to separate these highly similar genomes.

All the figures, tables and supplementary data were re-ordered according to the contents and logic order.

Metagenome comparison: I feel uncomfortable with the citrus root-associated soil microbiome (CRASM) merging the two distinct biomes of the citrus rhizosphere and its corresponding soils. I would keep and treat the rhizosphere and soil microbiomes separately. I expect the removal of citrus host reads (L118) to be different for the two sample types. I consider the identification of core functional traits for the citrus rhizosphere (compared to soil) appears the most important finding and advance of the manuscript (Fig. 6). To further highlight this, a side-by-side graph of soil vs. rhizosphere would be more appropriate than splitting the data by location (Figs. 6A&B), which is not discussed anyway in the results section.

Answer: The citrus originated reads were removed by mapping the reads to the three available high quality citrus genomes, and the filtered reads were used for de novo assembly. The results also demonstrated the rhizosphere samples contained more host originated reads than the corresponding bulk soil samples (Table S2). Recently, several metagenomics studies co-assembled the reads from multiple samples and found the assemblies showed better quality and reads utilization rate, and the assemblies were reliable as demonstrated by the following analyses (for example, Roux et al. Nature 537(7622), 2016; Bendall et al. ISME, 10, 1589–1601,2016; Lawson et al. Nature Communications 8, 15416,2017). In the present study, we also adopted the co-assembly approach using Megahit, a succinct de Bruijn graph based assembler which also includes reads mapping and low local coverage edge removal steps during the assembly to avoid mis-assembly, aiming to obtain better assemblies. Since the huge datasets (~1.5 Tb reads totally), we separated the 23 rhizosphere and 20 soil samples into 9 groups (Fig S2) based on their

microbial community similarity calculated by Mash, and assembled the pooled reads from each group using Megahit. The results demonstrated that this assembly approach generated much better assembly quality and higher reads utilization rates (Table S3). That's why we co-assembled the samples in this manuscript. Following your suggestions, we have conducted the comparisons with other microbiomes using the citrus rhizosphere and root microbiomes and bulk soil microbiomes separately. We compared the taxonomic and functional content between citrus-associated bulk soil and other soil microbiome, including desert and non-desert soil. We also compared citrus-associated bulk soil microbiome to free-living environments, such as ocean microbiome, and citrus rhizosphere microbiome to host-associated environments, such as human gut microbiome.

We removed the figure of the splitting the data by location and focused on the comparison between rhizosphere and bulk soil. Please see the new Figure 6 (C-G).

Terminology: The manuscript uses unnecessary complicated and sometimes confounding terminology. It is not immediately clear to what the authors refer to (e.g., does the expression 'root-associated soil microbiome' cover the 'rhizosphere', the 'root plus the rhizosphere', 'rhizosphere and soil'...?). I suggest to refer directly to the investigated sample types and simply term them as 'soil', 'rhizosphere' and 'root'. By the way, 'layer' is an uncommon term, 'compartment' is also widespread. I also felt that defining the many and similar abbreviations 'CRASM', 'CRAM' and 'CCRAM' does not make the manuscript easy accessible to readers. I do not see the point for introducing these abbreviations, especially also because the authors do not employ them in the discussion of their work. A general remark regarding abbreviations: define all abbreviations (e.g., PCoA, ITS, ARGs, ...) but only define abbreviations, which you often use in the MS (e.g., VPA is defined but not used).

Answer: Thank you for your suggestions. We revised these issues one by one as you suggested. The revised sentences are as follows: 'CRASM' was revised to 'global citrus-associated bulk soil and rhizosphere microbiome'; 'CRAM' was revised to 'global citrus rhizosphere and root microbiomes'; 'CCRAM' was revised to 'core citrus rhizosphere and root microbiomes'. And all the 'layer' were revised to 'compartment'. The general remark regarding abbreviations, such as PCoA, ITS and VPA were also defined.

Diversity analysis: I have a number of comments to the alpha and beta diversity analysis investigating the effects of sample type and location (L199...; Fig. 4A and B). The separate panels for sample type and location preclude to see possible interaction effects. I suggest to combine the two figures in one, were the three compartments are shown for each site. The logic order for sample type is 1) soil, 2) rhizosphere and 3) root. I do not see a decrease in Shannon diversity from soil to the rhizosphere (L203). The support for the claim in L204-207 is not easily documented as the distinction between Fig. 4B and S8 panel figures has to be made by reading the captions. Is Fig. 4B a mix of the sample types? What is about the fungal data? The 3 dimensions of the ordination graphs cannot be interpreted on 2D. It only makes sense to report PCo axes 1 and 2 in a printed graph and if helpful, axes 2 and 3 in a second 2D graph. I am

confused why the ordination graphs change between Fig. 4C and 4D. The samples points should stay in place in an unconstrained ordination and simply the color coding should change whether sample types or locations are highlighted. Please clarify if not eventually constrained ordination was applied. Fig. S9 is then redundant.

Answer: As you suggested, we re-analyzed the alpha and beta diversity. Please see the Figure 2 for 16S and Figure S7 for ITS. The alpha diversity (Shannon index) calculated from the amplicon-based data was significantly decreased from bulk soil or rhizosphere to root samples (corrected P-value <0.001, TukeyHSD, Fig 2B and Fig S7A). The old Fig. 4B is a mix of sample types and Fig S8 was separated by compartments.

The PcoA graphs were reshaped to 2D graph. The old Fig 4D was drawn based on the weighted Unifrac distance while Fig 4C was based on the unweighted unifrac distance, we corrected those results by using the unweighted unifrac distance and combined the compartments and location in one figure. Fig S9 was removed.

Kruskal-Wallis test: The authors utilize the Kruskal-Wallis test to identify location-specific taxa (L228...). This test is revealing, if there is any difference between the tested locations but does not identify which location(s) is/are different from each other. For the latter purpose, a dedicated non-parametric post-hoc test is required. Inspecting the Supplementary Data 3, none of the phyla has a significant adjusted P-value for soil and rhizosphere samples – hence the conclusion is that there are no differences between the locations!

Answer: We used other more reliable statistics methods for each comparison. The detailed information was described in the methods as follows:

1. Within-sample diversity was calculated for each sample using Shannon index based on the normalized OTU abundance table using rarefied method. The significant difference for alpha diversity across compartments were determined using two-way ANOVA and TukeyHSD method.
2. The citrus rhizosphere and root-specific taxonomic features were determined based on the abundance comparison between citrus and other plants using DESeq2 method^{68,70}. The reads count matrix for DESeq2 testing was normalized using DESeqVS method and the P-values were corrected using Benjamini-Hochberg (BH) method.
3. The significantly different features were determined based on the relative abundance comparison between citrus root-associated and other microbiomes using Wilcoxon signed-rank test¹⁴. The reads count matrix for Wilcoxon signed-rank test testing was normalized using TSS method and the P-value were corrected using BH method.
4. Based on the abundance profiles, the features (OTUs, genera, phyla and KOs) with significantly differential abundance across compartments were determined using the statistical method, such as DESeq2^{68,70} with a negative binomial generalized linear model. The reads count matrix for DESeq2 testing was normalized using DESeqVS method^{68,70}. For the rhizosphere enriched (abundance significantly higher than bulk soil) and depleted (abundance

significantly lower than bulk soil) genera and KOs detection in metagenomic data, paired DESeq2 comparison analysis was performed based on the reads count matrix of the genera and KOs across the bulk soil and rhizosphere samples. For the rhizosphere or root enriched and depleted OTUs and genera detection in 16S and ITS data, we also used the paired DESeq2 comparison analysis method to determine. P-values for multiple testing were corrected using BH method in DESeq2. All the items with corrected P-value below 0.05 were considered significant.

Based on the other reviewer's suggestions, we removed the comparison analyses between locations since this is out of the main focus of this manuscript. In the current study, we aimed to investigate the core citrus root associated microbiome, taxonomically and functionally, and to reveal whether there are citrus specific traits compared to other plants and ecosystems.

Balanced design: Without going to the method details, the reader gets the impression that the three sample types were collected at all 28 sites (L111). However, especially for soil (20 samples), the experimental design appears unbalanced. I could not square how Table S1 (23 locations) compares to the mentioned 28 sites in Fig. 1. Please clarify where samples are missing.

Answer: Thank you for your suggestions. We clarified this in methods, Fig 1 and Table S1. Totally we obtained 28 root samples, 23 rhizosphere soil samples (absence for the locations in Oman) and 20 bulk soil samples (absence for the locations in Oman and South Africa) from 28 representative locations in nine citrus producing countries, including USA, China, Brazil, Spain, Italy, Australia, France, South Africa and Oman, spanning the six continents where citrus grows (Fig 1A, Table S1). The bulk soil samples from South Africa were collected and DNA were extracted using the same kit (MoBio Powersoil DNA extraction kit) and the same extraction protocol as other sites did, however, the concentration and the total quantity of the DNA samples were too low to perform the library construction. We did the DNA extraction three times without success, thus did not include the bulk soil samples in the current manuscript. The collaborators in Oman only provided the 5 root DNA samples for this project.

Sampling depth: The CRASM has a good global sample coverage while the representativeness for the ocean and human gut metagenomes remains unclear. Can you place the 4x and 8x differences in gene coverage (L124) in scale to the ocean and human gut examples? How representative are the ocean and human gut metagenomes (worldwide and diverse sampling?). Along the same lines, I was wondering if the comparison for differences in gene coverage takes possible differences in sequencing depth into account (the deeper you sequence, the more new genes you will find).

Answer: The human gut (Li et al., 2014 Nature biotechnology) and ocean (Sunagawa et al., 2015 Science) metagenomes included more than 1200 samples across three continents and 240 samples across 68 locations representing all main oceanic regions (except for the Arctic) from three depth layers, respectively. Both of them are the large-scale and worldwide metagenomic sequencing. The metagenomic sequencing data sizes were amounting to 6.4 Tb and 7.2 Tb for

human gut and ocean microbiomes, respectively. Thus the the human gut and ocean samples are likely to be representative. In addition, the rarefaction analysis of detected genes showed that these two datasets are well represented. While the rarefaction analysis of unigenes of our gene catalogue indicated that it still did not reach a plateau (Fig 1D and Fig S3), suggesting high diversity and complexity within citrus-associated bulk soil and rhizosphere microbiomes.

Network analysis: People tend to over-interpret co-occurrence network graphs. The correlation analysis recapitulates simply the abundance behavior of microbes between two sample types: e.g., abundant taxa in soil positively correlate with each other and abundant rhizosphere taxa co-occur with each too, while negative correlation are mostly found between soil and rhizosphere-specific groups. This is not surprising, but this lies in the nature of the analysis with the correlation pattern reflecting the condition of the comparison (here: rhizosphere vs. soil). L297: explain, why the opposite abundance behavior of the mentioned taxa is intriguing. L302 and 305: refrain from interpreting ‘interaction’ or ‘communication’ between microbes as such correlation based analyses simply reveal co-occurrence patterns. In a way, co-occurrence networks can be seen as maps of abundance behavior for a certain condition (soil vs. rhizosphere).

Answer: Thank you for your suggestions. We revised this part as you suggested.

To reveal the potential relationship among core citrus rhizosphere and root microbiomes taxa, the co-occurrence networks on these habitat-specified core genera were generated based on strong ($R^2 \geq 0.7$) and significant ($P < 0.01$) correlations (Fig 4, Fig S11). Inside the network, most of the rhizosphere or root core nodes belong to Proteobacteria and harbor positive correlation with each other, whereas most of the rhizosphere or root depleted nodes are from Cyanobacteria and Actinobacteria (Fig 4, Fig S11). Between the enriched and depleted network, the majority of the correlations were negative (Fig 4, Fig S11). Furthermore, we found that the rhizosphere or root core nodes positively associated more among themselves than with the rhizosphere or root depleted nodes (Fisher exact test; $P=0.03$) (Fig 4, Fig S11). These results suggest that the individual core microbes in the rhizosphere or root habitat tend to associate with other core members, probably through their similar micro-environmental cues, such as the concentrations of root exudates, or via their complementary roles in the biogeochemical cycles. The majority of the connections between the rhizosphere or root core nodes and rhizosphere or root depleted nodes are negative, further suggesting that the micro-environmental factors reshaped the rhizosphere or root core microbes from the corresponding bulk soil microbial community.

Graphic help: In general, it would be a service to the reader if the type of compartment (soil, rhizosphere or root) and type of data (amplicon, metagenome) is indicated in all graphs and there is no need to retrieve this information from the caption

Answer: Thank you for your suggestions. We revised the graphs per your suggestions. We added the data type and type of compartment for the figures which are easily confused.

Code: all bioinformatic code must be made publically available!

Answer: We mainly used the published programs to deal with the data and made some scripts to draw figure. We have posted the custom scripts and the main commands for our data analysis on

our data website.

Minor comments and suggestions:

- The manuscript contains numerous language ‘bumps’ and typos. E.g. in L42, “The plant root-associated plays...” or “Plant root-associated microbiomes play_...”. Similar in Lines 60, 57, 68, 128...)

Answer: Thanks. We have read the manuscript multiple times and revised them one by one.

- L49: You refer to the human gut, no?

Yes, human gut. And this sentence was removed as suggested by the other reviewer.

- L50: Avoid unexplained methodological information (identity and coverage).

Answer: Thanks. This sentence was removed as suggested by the other reviewer.

- L54: Rephrase: “...the amplicon of prokaryotic and fungal ribosomal operon marker genes.”

Answer: Thanks, we revised it.

- L62: To which soil properties do the authors refer to?

Answer: Thanks. The soil properties included soil nutrition, pH, minerals and structures. This sentence was removed as suggested by the other reviewer.

- L68: Avoid dual mentioning.

Answer: Thanks, we removed it.

- L77: The authors report that the PCR-based taxonomic profiling is similar with the unbiased shotgun metagenomic taxonomy. I was wondering, if it is necessary here to raise the possible issue of PCR biases of amplicon approaches? Add references or examples for the ‘PCR bias’ and ‘niche-specified’ claims.

Answer: Thanks, we removed it as you suggested.

- Table S2: typo in the header (“sickle treated sington”), which samples are rhizosphere and soils?

Answer: We revised them. See Table S2.

- Fig. 1C: I greatly appreciate the metagenome comparison of the citrus rhizosphere and corresponding soil to ocean and human guts. Why is human gut missing in this figure?

Answer: We added the human gut comparison as demonstrated in Fig S14.

- Fig. 1E: I do not see the need for this figure as this message (L127) is part of Fig. 1C and because annotation differences between locations were not statistically investigated or reported.

Answer: Thanks. We removed it.

- Fig. 2A: Use one color with different hues for the various classes of Proteobacteria. This will help the reader to appreciate the Proteobacteria claim in L139 more easily. I could not find a table that supports the statistic analysis. I suggest to add an asterisks to the taxon name in the legend for the taxa that are significantly different between the three biomes. Update the Fig. 2C in a similar manner with the statistic information.

Answer: Thanks. We revised as you suggested. The table for the statistics analysis was added.

- Fig. 2B: I would use more intuitive colors: blue for the ocean and green or yellow for citrus.

Answer: Thanks. We revised accordingly.

- Fig. 2E/Fig. S4: Ordination graphs would be more straight-forward to read.

Answer: Thanks. We revised them. Please see the Fig S15 and S20.

- L149-151: add reference to Fig. 2D.

Answer: Thanks. We added it.

- L181: While the authors mention the average sampling intensity for the metagenomes, this is not done here for the amplicon approach.

Answer: Thanks. We added it. A total of more than 2.12 and 1.34 millions high quality tags were generated for 71 and 62 16S V4 and ITS2 sequencing samples, respectively. After removal of citrus host originated tags, on average 21942, 22797 and 9922 effective 16S V4 tags, and 21523, 22555 and 19882 effective ITS2 tags were generated for bulk soil, rhizosphere and root samples, respectively.

- L182: Explain how the metagenomic data was used to assist with the taxonomic analysis.

Answer: This sentence was not necessary. We removed it. We analyzed the amplicon-based data (including 16S and ITS profiling), and metagenomic unigenes integratively whenever possible.

- L249: Acidobacteria were markedly lower in abundance compared to Proteobacteria and Actinobacteria (Fig. 3A) and I was wondering if the authors can explain, why most of the MAGs belonged to Acidobacteria? Differences in genome size, less complex assembly, fewer repeats, ...?

Answer: The purpose of MAG analysis was to provide cues for the bulk soil to rhizosphere enrichment process of the citrus root-associated microbiome at single genome level, and the results were consistent with those from the community-based analyses. However, we went through the manuscript carefully based on the all reviewers' suggestions, and thought the MAG section was out of logic, did not provide enough novel insights and information for this manuscript, and have removed the MAG analysis in the current manuscript. In addition, we found that these MAGs represented a very small part of our data. For example, the total sequence length of all MAGs, number of predicted genes, average percentage of mapped reads were only 1.3 Gb (10 % of total sequence length for metagenomic contigs), 1.42 millions (0.6 % of 230 million metagenes) and 2.6%, respectively. Furthermore, the abundant Proteobacteria and Actinobacteria included too many similar and high abundant genomes. Based on the current methods using the tetra-nucleotide frequencies, abundance and GC contents, it's hard to separate these highly similar genomes.

- L252: I am surprised that 42 (10%!) of the MAGs could not be assigned even at highest taxonomic phylum rank. Can you exclude technical artifacts?

Answer: The MAG data have been deleted as reasoned above.

For the previous studies from aquifer systems, the authors defined 47 putative phyla based on the 554 MAGs (More than 42% of 1297 MAGs). In the more complex soil microbiome, this percentage may be normal or even underrepresented.

• L276: what groups? The sample types?

Answer: Yes, sample groups based on compartments.

• L289: Fungal data not inspected?

Answer: Thanks. We added the fungal data as suggested. In addition, 7 and 6 core rhizosphere and root fungal genera were also identified using the same method, respectively (Supplemental data 3). Some of these core fungal genera, such as *Rhizophagus* and *Glomus* are beneficial to plants.

• L290: I would be very careful with the 'beneficial' claim. *Pseudomonads* as well as *Agrobacteria* comprise well-known plant pathogens. The metagenome further contains *Ralstonia* or *Xanthomonas*!

Answer: Thanks. We mentioned more carefully as you suggested. Some of them may be the beneficial bacteria.

• Fig. 5A/B: What is the taxonomic/sequences-based overlap/agreement between 16S and metagenome data?

Answer: The core rhizosphere genera generated by metagenomic data included more than 63% of the core root genera generated by 16S data.

• Fig. 5C/D: Explain the scale (negative values) for the relative abundance. Too many colors to be discriminated. Label the locations with the previously used abbreviations (Aus, Br,...), I would also label the sample types without a color code. I guess the left-hand color bar should refer to phylum level taxonomy? Legend is missing. The same comments apply to Figures 6E and F.

Answer: Thanks. We re-drew the figures. To show more clear pattern, we dealt with the data using the row normalization of the relative abundance for each genus using Pheatmap packages in R. Scale in this package means removing the mean (centering) and dividing by the standard deviation (scaling). This is actually Z-score value.

• L292: I am confused how the MAGs are linked with the CCRAM?

Answer: Thanks. We revised it. The MAG data was removed as reasoned above.

• L363: With regard to microbe-microbe interaction, any indication of type-6-secretion system increase in the rhizosphere?

Answer: Yes, please see the Fig S21B.

• L383-387: This claim is strongly linked to sampling intensity. The authors have 16S data with a sampling depth of 20ish thousand reads per sample, which is rather low for soil profiling.

Answer: We removed this claim.

• L398: Rephrase to "Citrus is planted worldwide and subjected to nutrient limitations ..."

Answer: Thanks. We revised it.

• L407: It remains unclear how the citrus microbiome data could be deployed for synthetic microbiome engineering. Examples or ideas are needed. In particular, how should network analysis help to predict inoculation success of microbes to agro-ecosystems?

Answer: This is a good point. Synthetic microbiome engineering is at the early stage of application. The core citrus microbiomes might serve as a starting point for synthetic microbiome engineering application as it is almost impossible to reapply all the microbes in the community. In our previous works, we introduced the species from *Bradyrhizobium* and *Burkholderia* to citrus plant root systems, which improved the citrus growth and health (Zhang et al., 2017; Riera et al., 2017). Based on the co-networks of these core microbiomes, we can combine different strains which have the synergistic effect and introduce the mixed microbiomes into the plant root system. Recently, some researchers designed the synthetic bacterial communities to predict plant phenotypes according to their relationship (Paredes et al., 2018).

• L525: Explain how the reads were “cleaned” at BGI?

Answer: Thanks. We revised it. The raw reads from metagenome sequencing were used to generate clean reads by adaptor sequences removing, low quality reads (reads with ‘N’ base and minimum quality threshold was 20) trimming and removing at BGI-Shenzhen, China.

• L566: 23 rhizosphere and 20 soils samples

Answer: Thanks. We revised it.

Reviewer 2

The manuscript “The structure and function of the global citrus root-associated microbiome” by Xu, et. al. attempts to characterize the taxonomic and functional components of the citrus rhizosphere microbiome through the use of metagenomic analysis. This manuscript represents a considerable portion of work and is based on a rather a large collection of citrus rhizosphere samples from throughout the world. However, there are several issues concerning the writing and interpretation of the results could use extensive revision.

My concerns can be listed as following:

- 1.) Focusing on microbiome functional over-lap between the plant rhizosphere with ocean and human gut micro-environments is inappropriate.
- 2.) A lack of description of the germplasm and soil characteristics in samples used in this study.
- 3.) A lack of discussion regarding the disadvantages and assumptions that are inherent in metagenomic studies of the microbiome

The author’s make dubious claim that the human gut is functionally analogous to the rhizosphere and support the claim with a single review article. However, there are multiple published empirical studies that support that fact that very few microbial taxa overlap between these environments (see Thompson, Nature, 2017 for a recent example) suggesting that there should be very little functional overlap in the metagenomes of these environments.

Additionally, the human gut a relatively consistent environment with acid pH and temperature

and is far more resilient upon perturbation with diet (David et. al, Nature, 2014) where as the authors freely admit that the rhizosphere is far more variable and is more strongly influenced by factors such as “geographic locations, salinity, temperature, oxygen, nutrients, pH, day length, and diseases^{20,23–30}” which influences microbial community composition and function. The authors also make no justification why they compared rhizosphere metagenomic data with oceans, which is also widely accepted as being drastically different in microbial composition. Thus, the mere fact that metagenomic analysis of citrus rhizosphere’s differs from human and ocean environments is neither surprising nor novel. The manuscript would be better served by focusing on the advantages and disadvantages of metagenomic analysis for understanding community function and that this paper focuses on comparing a plant rhizosphere community (which has both free- living and host-associated components) with two well-characterized metagenomes (one free- living, i.e. ocean dataset, and one host-associated, i.e. human gut).

Answer: Thank you for your valuable suggestions. The primary aim of this manuscript, as the first global-scale plant root-associated microbiome research, is to discover the gene resource and identify the compartment-enriched microbes and the functional traits of the citrus root associated microbiomes, aiming to contribute to the understanding of plant root-associated microbiomes. Following your valuable suggestions, we removed the claims and discussion for microbiome taxonomic and functional overlaps between the plant rhizosphere with ocean and human gut micro-environments. Then we sought to reveal the microbiome differences between citrus rhizosphere and human gut, and between citrus-associated bulk soil and ocean. In addition, as you suggested, we did the comparison among the microbiomes from different soil systems, including citrus-associated bulk soil, desert and non-desert soil.

Per your suggestions and reviewer 1’s suggestions, we re-organized the manuscript as following. In the current version, we first showed the results of “Global citrus-associated bulk soil and rhizosphere microbiome gene catalog construction”, then presented the “Taxonomic content in global citrus rhizosphere and root microbiomes”, defined the “Core taxa of the citrus rhizosphere and root microbiomes”; next, we reported the “Comparison of rhizosphere and root microbiomes between citrus and other plants”, as well as the “Taxonomic content comparison among microbiomes of citrus-associated bulk soil, rhizosphere and other ecosystems, including non-citrus associated soil, human gut and ocean samples”. Finally, we performed the “Functional composition of the citrus-associated bulk soil, rhizosphere microbiomes and comparison with that from other ecosystems” and also defined the “Core functional traits in global citrus rhizosphere microbiome”.

The initial purpose of MAG analysis was to provide cues for the bulk soil to rhizosphere enrichment process of the citrus root-associated microbiome at single genome level. The results were consistent with those from the community-based analyses, but these MAGs represented a very small section of our data, after carefully thinking all reviewers’ suggestions, we agree that the MAG section did not provide enough novel insights and information for this manuscript and have removed the MAG analysis in the current manuscript

to make it concise.

The authors also do not adequately describe the level of natural variation in the citrus host used for this study within the manuscript and have hidden this information in the supplemental tables. Presumably, these are all agricultural fields that are using a narrow subset of domesticated citrus varieties grown on cultivated plots. Citrus is well-known to produce a wide of terpenoids that varies widely both qualitatively and quantitatively among both domesticated and natural citrus varieties. Therefore, it is unclear if some of the rhizosphere enrichment observed in the analysis is common to a wide variety of citrus genotypes or a function of only a single citrus variety that is widely cultivated throughout the world. Additionally, it is unclear if the geographic differences highlighted in the results are due to abiotic soil conditions at that site or plant varietal differences for citrus that is commonly cultivated within that region of the world. This lack of detail also obfuscates the general utility and extensibility of this analysis for both citrus and other crop species.

Answer: Thank you for your suggestions. We added the description of the citrus host used in this study (Supplementary data 1 and Table S1). The 28 locations included different soil type (~ 7 types), climate type (~ 8 types), pH (5.2 to 8.3), contents of organic C, N and P, and diverse citrus germplasms (~ 13), which will guarantee the high diversity and representation for our global samples. Furthermore, to guarantee the reliability for the enriched analysis, we performed the enrichment analysis from bulk soil to rhizosphere or to root compartment based on the pair wise comparison (each sample site includes the bulk soil, rhizosphere or root samples from the same trees and location) using DESeq2 method. Based on this method and high diversity samples, we found the common enriched taxa and function from different citrus germplasms and different locations.

The natural variation and citrus germplasm for each location was significantly different. As you suggested, the differences were not clear. We removed the comparison results among the geographic locations. We aimed to unearth the core citrus root associated microbiome, taxonomically and functionally, and to reveal whether there are citrus specific traits compared to other plants and ecosystems.

Lastly, both the introduction and discussion lack any consideration of the potential limitations of the method in addition to advantages. Since metagenomic analysis relies on the enrichment of DNA in a particular environment, results often strongly correlate with taxonomic enrichment regardless of the transcriptional/translational state of the gene in question. For example, cephalosporin-biosynthesis identified in this manuscript is common among proteo- and actinobacteria and thus should be enriched in the rhizosphere environment. However, this genomic enrichment does not mean that these pathways are expressed and utilized under within this micro-environment and also cannot be used to determine if these are used in either the biosynthesis or degradation of cephalosporin-like secondary metabolites. This criticism of the technique applies to interpretation of all results from genomic enrichment of biochemical pathways at the DNA level. The authors need to either: 1.) control for taxonomic enrichment in this analysis, or 2.) address the limitations of the method in either the introduction and/or the discussion.

Answer: Thank you for good suggestions, As you suggested, we discussed the potential limitations of the method based on metagenomic data and future direction in the discussion. The revised contents are as following.

“Overall, this study provides the largest gene catalog so far which will provide the genomic resource for the scientific community. In addition, understanding the global citrus microbiome lays the foundation for future agriculture practices through the microbiome engineering approach⁶³. However, this is still at very early stage for soil- and plant-associated microbiome study. Firstly, the current metagenomic data only give us the limited information and very basic knowledge for the highly complex soil- and plant-associated microbiomes, which is demonstrated by the unsaturated rarefaction of gene number for the citrus-associated bulk soil and rhizosphere microbiome gene catalog. Another limitation of the metagenomic data is that it could not reveal whether the genes detected or enriched are expressed or functional at the tested conditions. Thus, the findings generated from the metagenomic data need to be supplemented by other data and experiments. In the future, we need more large-scale and multi-omics data, such as meta-transcriptome, meta-proteome and meta-metabolome, as well as conducting culture-dependent experiments and multi-omics analysis to further improve and enhance our understanding of the soil and root associated microbiome. ”

Detailed Comments:

Abstract:

42: *may* play roles Not under all conditions and are likely small factors relative to larger trends.

Answer: Revised, thanks

43-44: “... are largely unknown.” The taxonomic contents of plant rhizospheres are fairly well characterized, as cited by the authors.

Answer: We removed the taxonomic contents. Thanks

47-52: Why are the authors comparing their metagenome analysis to human and oceans? I’m sure that there are other previous soil metagenomic papers to compare their results to (e.g. Fierer 2012 PNAS). The scope of comparisons is inappropriately wide. See above.

Answer: Thank you for your suggestions. We removed the claims and discussion for microbiome taxonomic and functional overlaps between the plant rhizosphere with ocean and human gut micro-environments. We did keep some analysis regarding microbiome differences between citrus rhizosphere and human gut, and between citrus-associated bulk soil and ocean. We added the analysis with previous soil metagenomic data.

55: Is the core root-associated taxa specific to citrus or have the authors just re-identified rhizosphere specific taxa.

Answer: We identified rhizosphere specific taxa based on the multi-citrus germplasm and locations. And some of them are specifically enriched in citrus when compared with other plants.

Introduction:

The whole first paragraph could be removed without much impact on the remainder of the story. It jumps around in focus and does not present the following work well.

Answer: Thanks, we removed this part as you suggested.

59: The animal gut is not widely considered to be analogous to the soil rhizosphere. See above.

Answer: We removed it . Thanks.

63-67: This is more of a conclusion and potential broader impacts statement.

Answer: We removed it. Thanks.

75-79: The authors comment on the limitations of 16S DNA profiling, but fail to mention the limitations in metagenomic analyses, such as KO-enrichment closely tracks with microbial phylogeny and should not be extrapolated to believe that enriched genetic signatures at the DNA level are functional in that environment. See above.

Answer: Thank you for your suggestions. We added the following in the discussion: Another limitation of the metagenomic data is that it could not reveal whether the genes detected or enriched are expressed or functional at the tested conditions.

87-93: This should be the beginning of the introduction. The focus of the study is on citrus and NOT on plants in general.

Answer: Thank you for your suggestions. We revised it.

Methods

The sample numbers don't always add up and/or it is unclear how and when the authors attempted to pool. For example:

448: The authors list 71 samples split among roots, rhizosphere, and bulk samples, but previously stated that they sampled 4 trees from 28 sites across the globe, adds up to 112 samples not including the bulk soil. Please clarify.

Answer: Thank you for your suggestions. We revised it. Four healthy citrus trees were selected from each selected grove, and the roots, rhizosphere and the corresponding bulk soil samples were collected from each tree. The roots, rhizosphere and the corresponding bulk soil samples of four trees from same grove were pooled together as one sample for each location, respectively. Totally we obtained 28 root samples, 23 rhizosphere soil samples (absence for the locations in Oman) and 20 bulk soil samples (absence for the locations in Oman and South Africa) from 28 representative locations in nine citrus producing countries, including USA, China, Brazil, Spain, Italy, Australia, France, South Africa and Oman, spanning the six continents where citrus grows (Fig 1A, Table S1).

528-529: It is unclear what samples are within the 43 samples listed for metagenomic analysis. Presumably, it refers to the rhizosphere and bulk samples. Please make explicit.

Answer: 43 samples included 20 bulk soil and 23 rhizosphere samples. Detailed information was listed in Table S2

Also:

445-447: Please expand on the bulk soil site identification and collection.

Answer: We revised it. Thanks. The soil from depth 5 cm-15 cm without any roots were collected from multiple sites near the selected trees and stored as the bulk soil sample at 4°C until DNA extraction on the same day.

452-489: Place in supplement and only restrict the deviations from the manufactures protocol for the main text.

Answer: This was a standard manufactures protocol. We removed it. Thanks.

496-498: The authors MUST expand on the how the DNA libraries for the metagenomic sequencing was made. A part of a sentence is wholly inadequate.

Answer: We revised it. Thanks. For the amplicon library preparation, the amplification of 16S and ITS DNA fragments were performed using the previously described common amplified primers and methods of prokaryotic 16S rDNA V4 region (515F and 806R)^{69,70} and fungi ITS271. For the metagenomic library preparation, the metagenomic DNA was sonicated to 350-base pair (bp) size range. DNA fragments were then end repaired, 3'-adenylated and amplified using Illumina sequencing adapter-specific primers. After quality-control, quantification and normalization for DNA libraries, 150 and 250bp paired-end reads were generated from the Illumina HiSeq4000 and MiSeq platform under the modified manufacture's instruction for metagenome and amplicon (16S and ITS), respectively.

508: I would suggest the authors use a denoising focused pipeline, such as uNoise3 or DADA2, to remove sequencing errors from their 16S analysis.

Answer: Thank you for your suggestions. The UPARSE pipeline also included the sequencing errors corrected steps. For our purpose, this pipeline is sufficient. In the future, we will try these methods.

510: Please cite the reference databases used.

Answer: We added them. Thanks

513: Please cite the TSS normalization method used.

Answer: We added it. Thanks

514: the Shannon index should only be calculated on rarefied data since more species will be identified depending on sequencing depth and normalization can skew presence/absence data used to calculate alpha-diversity.

Answer: Thank you very much! We re-analyzed it based on the rarefied data (using the minimal read count). Within-sample diversity was calculated for each sample using Shannon index based on the normalized OTU abundance table using rarefied method. The significant difference for alpha diversity across compartments were determined using two-way ANOVA and TukeyHSD method.

542, 566-567, 576-577: There doesn't seem to be any normalization method for the metagenome analysis. Thus, enrichment analyses could very well be the function of sequencing depth artifacts. In addition, I don't see how the authors controlled for multiple testing? The results mention FDR corrections, but should be described here.

Answer: Thanks. We revised it and described the detailed information for these parts as

following.

(1) The relative abundance tables for taxa (OTU, genus and phylum) were generated based on the reads count for each taxon across sample using total-sum scaling (TSS) method. Within-sample diversity was calculated for each sample using Shannon index based on the normalized OTU abundance table using rarefied method. The significant difference for alpha diversity across compartments were determined using two-way ANOVA and TukeyHSD method.

(2) The citrus rhizosphere and root-specific taxonomic features were determined based on the abundance comparison between citrus and other plants using DESeq2 method^{68,70}. The reads count matrix for DESeq2 testing was normalized using DESeqVS method and the P-value were corrected using Benjamini-Hochberg (BH) method.

(3) The significant different features were determined based on the relative abundance comparison between citrus root-associated and other microbiomes using Wilcoxon signed-rank test¹⁴. The reads count matrix for Wilcoxon signed-rank test testing was normalized using TSS method and the P-value were corrected using the BH method.

(4) Based on the abundance profiles, the features (OTUs, genera, phyla and KOs) with significantly differential abundance across compartments were determined using the statistical method, such as DESeq2^{68,70} with a negative binomial generalized linear model. The reads count matrix for DESeq2 testing was normalized using DESeqVS method^{68,70}. For the rhizosphere enriched (abundance significantly higher than bulk soil) and depleted (abundance significantly lower than bulk soil) genera and KOs detection in metagenomic data, paired DESeq2 comparison analysis was performed based on the reads count matrix of the genera and KOs across the bulk soil and rhizosphere samples. For the rhizosphere or root enriched and depleted OTUs and genera detection in 16S and ITS data, we also used the paired DESeq2 comparison analysis method to determine. P-values for multiple testing were corrected using BH method in DESeq2. All the items with corrected P-value below 0.05 were considered significant.

582: I would recommend running the permutation 1,000x. Thresholds can be arbitrarily low with inadequate replication in permutation estimation.

Answer: Yes, we used 1000X. It was typo. This part related to MGA has been removed.

595-596: A threshold of $r \geq 0.7$ seems extremely low and only translates to a nominal R^2 of 0.49. This likely results in excessively connected network for the given data.

Answer: The threshold is $R^2 \geq 0.7$. We revised it. Thanks.

Results

Figures are often unnecessarily complex or not properly referred to in the text.

Answer: Thanks. We revised accordingly.

Fig. 1C: out of order relative to the text and lacks an analysis of the human gut. Also, since the rarefaction curve suggests that the metagenome is under-sampled, the proportion of reads assigned to viruses and eukaryotes/protists for the citrus rhizosphere is not likely representative. Also, the authors should define how they are describing the last universal

common ancestor (LUCA) in this context. Move to supplement or remove completely.

Answer: Thank you for your suggestion. We re-drew this figure, which only showed the percentage of bacteria & archaea and unknown, and added the analysis of human gut. Please see the new figure S14.

Fig. 1D: By including the combined 'All' unigene category in the graph obfuscates the fact that neither the rhizosphere nor the bulk soil are close to saturating the potential unigenes for the microbial community. While the authors do state this, the graph is wholly misleading as is.

Answer: Thanks. We re-drew this figure separately. Please see Fig 1D and Fig S3.

Fig. 1E: would be better described as a table rather than a graph. Move to supplement.

Answer: Thanks. We removed this figure per your and another reviewer's suggestions.

127: needs to state the variance in unknown genes across the sites sampled.

Answer: Thanks. We removed the results for location comparison due to the reason described above.

133-137: Analysis out of place. Also these results are from the metagenomic analysis and not the 16S profiling. The remainder of the paragraph is not clear if the taxonomic profiling is coming from the 16S sequencing or from a phylogenetic analysis of the metagenomic assembled genomes.

Answer: Thanks. We reorganized the structure of the manuscript. This paragraph is based on the metagenomic data.

Fig 2A&B: please label if these are from 16S profiling and/or the metagenomic analysis.

Answer: We revised it. Thanks. Please see the new figure Fig 5 A&B.

144: This is not necessarily noteworthy as this has been shown consistently. Most recently by Thompson et. al. Nature (2017)

Answer: We removed it. Thanks.

149-174: Please also compare these results with other soil metagenomic analyses, such as Fierer et. al. PNAS 2012. The bulk soil results should compare directly.

Answer: Thanks. We conducted the analyses as you suggested.

151: "The KOs..." the CRASM KOs?

Answer: The KOs for all the data sets.

154-155: Speculative and obvious.

Answer: We revised it. Thanks.

156-159: Difficult to follow and a run-on sentence. Split up.

Answer: We revised it. Thanks. More than 95% of KOs in human gut microbiome and more than 94% of KOs in ocean microbiome were present in citrus-associated bulk soil and rhizosphere microbiomes. Meanwhile, more than 31% of KOs identified in citrus-associated bulk soil and rhizosphere microbiomes, which are mainly involved in signaling and cellular processes (44.8% of the citrus-associated bulk soil and rhizosphere microbiome specific KOs),

metabolism (28.9%) and genetic information processing (22.6%), were absent in the other two niches (Fig S19B).

Fig 2E: move to supplement

Answer: We revised it. Thanks. Fig 2E was re-drawn as Fig S15B with PCA plot.

170-174: This is a ludicrous claim that the soil is the source of inoculum for oceans and the human gut and doesn't follow any known ecology or evolutionary theory. Especially for humans guts with individuals from developed nations where there is a long production line of food washing/processing/inspection/storage between the soil and human consumption. More likely, these are highly generalist taxa that are highly plastic in their ecological niche and are extremely ubiquitous in all microenvironments, including the human gut and oceans.

Answer: Thank you for your suggestions. We removed this claim.

175-179: It is difficult to parse the difference between the goals of the citrus root associated soil microbiome (CRASM) and the citrus root associated microbiome (CRAM). Please use a different description and acronym.

Answer: We revised these issues as you suggested. The revised sentences are as follows: 'CRASM' was revised to 'global citrus-associated bulk soil and rhizosphere microbiome'; 'CRAM' was revised to 'global citrus rhizosphere and root microbiomes'; 'CCRAM' was revised to 'core citrus rhizosphere and root microbiomes'.

184: "amplicon"? do you mean 16S profiling?

Answer: included 16S and ITS profiling.

186-187: Please use the coefficient of correlation (r^2) for Spearman correlations instead of the coefficient of determination (r). Use of r is misleading and gives the impression of a stronger relationship than actually exists.

Answer: We revised it. Thanks.

191-216: Figure order is confusing. The authors are asking the reader to jump around the figures often and is making it difficult to follow the logic. I would suggest restructuring the figures according to logical flow of the claims rather than by similar analysis.

Answer: Thanks. We re-organized the structure of the manuscript and the order of figures as described above.

204: Given the complex influence of microcosm and location on diversity, Kruskal-Wallis is the wrong test. I would suggest at least a two-factor linear model with passion link function for Shannon Index or equivalent non-parametric test. This should more closely correspond to the VPA results. Also, this should be done on rarefied data, not TSS transformed data. See comments on methods.

Answer: Thanks. We re-analyzed them. The normalized OTU abundance table was generated based on the reads count for each OTU across samples using total-sum scaling (TSS) and rarefied methods. Within-sample diversity was calculated for each sample using Shannon index based on the normalized OTU abundance table using rarefied method. The significant difference for alpha diversity across compartments were determined using two-way ANOVA

and TukeyHSD methods.

213-239: Same issue as above. The authors need a more flexible statistical framework. Also, they should incorporate a multiple testing correction to reduce their false-positive rate.

Answer: We removed the comparison among locations, due to the reason described above. We re-analyzed all the comparisons using the more flexible statistical framework as previous studies per your suggestions.

228-229: How do these site-specific signatures correspond with the soil characteristics from each site? For instance, Brazilian soils are often contaminated with high levels of aluminum, which can limit calcium availability in the soil community. This is especially important as many studies have shown that soil abiotic conditions are more important for microbiome taxonomic composition.

Answer: We removed the comparison among locations, due to the reason described above.

242-246: What is not clear to me is if the MAGs were constructed within site or across the entire set. If the later, what assurances do we have that the in silico constructed MAGs are not just a bioinformatic artifact? To address this, the authors should state what percentage of the reads for a MAG are from the largest contributing site (cis- reads) vs reads that came from other sites (trans- reads).

Answer: Thank you for your suggestions. We have removed the MAG data as reasoned below. The purpose of MAG analysis was to provide cues for the bulk soil to rhizosphere enrichment process of the citrus root-associated microbiome at single genome level, and the results were consistent with those from the community-based analyses. However, we went through the manuscript carefully based on the all reviewer's suggestions, and thought the MAG section was out of logic, did not provide enough novel insights and information for this manuscript, and have removed the MAG analysis in the current manuscript. In addition, we found that these MAGs represented a very small part of our data. For example, the total sequence length of all MAGs, number of predicted genes, average percentage of mapped reads were only 1.3 Gb (10 % of total sequence length for metagenomic contigs), 1.42 millions (0.6 % of 230 million metagenes) and 2.6%, respectively. Furthermore, the abundant Proteobacteria and Actinobacteria included too many similar and high abundant genomes. Based on the current methods using the tetra-nucleotide frequencies, abundance and GC contents, it's hard to separate these highly similar genomes.

266: "...higher *taxonomic* resolution...."

Answer: We revised it. Thanks.

262-273: This should be moved up in the results to justify that these results are similar to what has been seen for other model systems. The authors should also mention if there are any taxa that appear to be qualitatively specific to the field citrus rhizosphere. If not, please state explicitly.

Answer: We revised and added them. Thanks. Furthermore, 30 and 63 genera were specifically enriched in citrus rhizosphere and root microbiomes when compared with other plants, respectively. 18 citrus rhizosphere or root specifically enriched genera belonged to the core citrus rhizosphere or root microbiomes, such as *Cupriavidus*, *Ochrobactrum*,

Pseudomonas, *Sphingobium*, *Dyella*, *Hylemonella*, *Sinorhizobium*, *Stenotrophomonas*, *Bradyrhizobium* and *Sphingomonas* (Table S6 & S7, Supplemental data 4 & 5). The identified common abundant plant root-associated and citrus root-associated specifically enriched taxa may serve as the backbone components of plant and citrus microbiomes, respectively.

274: Please don't use this acronym (CCRAM). It's just confusing with CRAM and CRASM and it makes it hard for the reader to keep follow what community is under discussion. Just leave it as the core rhizosphere community.

Answer: We have removed the three acronyms altogether. Thanks. The revised sentences are as follows: 'CRASM' was revised to 'global citrus-associated bulk soil and rhizosphere microbiome'; 'CRAM' was revised to 'global citrus rhizosphere and root microbiomes'; 'CCRAM' was revised to 'core citrus rhizosphere and root microbiomes'.

Fig 5A and B: Please label the Venn diagram with compartment/method.

Answer: We revised it. Thanks. Please see the new figure Fig 5.

Fig 5C and D: Please label the top and bottom of the heat plot with names instead of colors as this is difficult to follow and is confusing for a color-blind audience.

Answer: We revised it. Thanks. Please see the new figure Fig 5.

285-290: Are these core taxa common members of core communities for other plant species or are these specific to citrus?

Answer: We identified rhizosphere specific taxa based on the multi-citrus germplams and locations. And some of them are specifically enriched in citrus when compared with other plants.

291: Is this relative to the 16S data or the metagenomic data?

Answer: We did both for 16S data (new Fig S11) and metagenomic data (new Fig 4).

292: Again, please use r^2 and not r for correlations.

Answer: We revised it. Thanks.

294: Network robustness is not a function of the significance of the edges. It is the ability of the network to maintain it's topology, structure, and/or output based on perturbation or removal of individual nodes and/or edges. Please remove as this statement is not supported by the data.

Answer: We revised it. Thanks.

302: "...interact..." change to "associate"

Answer: We revised it. Thanks.

305-309: The authors make extremely broad assumptions to conclude that core microbes "... tend to communicate with other core members, probably through their metabolic handoffs. ..." from a simple correlational relationship. They could also just respond similarly to environmental cues, e.g. increasing root exudate concentrations, which would also result in a significant correlation.

Answer: Thanks. We revised it as you suggested as shown below.

“To reveal the potential relationship among core citrus rhizosphere and root microbiome taxa, the co-occurrence networks on these habitat-specified core genera were generated based on strong ($R^2 \geq 0.7$) and significant ($P < 0.01$) correlations (Fig 4, Fig S11). Inside the network, most of the rhizosphere or root core nodes belong to *Proteobacteria* and harbor positive correlation with each other, whereas most of the rhizosphere or root depleted nodes are from *Cyanobacteria* and *Actinobacteria* (Fig 4, Fig S11). Between the enriched and depleted network, the majority of the correlations were negative (Fig 4, Fig S11). Furthermore, we found that the rhizosphere or root core nodes positively associated more among themselves than with the rhizosphere or root depleted nodes (Fisher exact test; $P=0.03$) (Fig 4, Fig S11). These results suggest that the individual core microbes in the rhizosphere or root habitat tend to associate with other core members, probably through their similar micro-environmental cues, such as the concentrations of root exudates, or via their complementary roles in the biogeochemical cycles. The majority of the connections between the rhizosphere or root core nodes and rhizosphere or root depleted nodes are negative, further suggesting that the micro-environmental factors reshaped the rhizosphere or root core microbes from the corresponding bulk soil microbial community.”

Fig 5E: Please label the left and right side of the dotted line. Also, use a different color for negative correlations. Also, the authors did not filter the network for the same correlations, i.e. the correlation of A to B is the same as B to A. This is most obvious with the singleton correlations, such as *Caldisericum* with *Planktothricoides* and *Pseudoduganella* with *Duganella*. Please filter.

Answer: We revised it. Thanks. Please see the new figure Fig 4.

Fig 6A and B: uninformative, move to supplement.

Answer: We removed it. Thanks.

Fig 6C and D: Again, please label the Venn diagram. Present in what? Enriched in what?

Answer: We revised it. Thanks. Please see the new figure Fig 6.

Fig 6E and F: Please label the sides and do not use the color system. Also, please split up the legend because it is confusing to relate compartment with site.

Answer: We revised it. Thanks. Please see the new figure Fig 6.

Fig 6G: Again, please filter the network edges for forward and reverse relationships and use a different color for negative correlations. Also, please label the subnetworks that are being highlighted.

Answer: We revised it. Thanks. Please see the new figure Fig 6.

325-327: Please expand on why metabolic pathways, such as carbohydrate and amino acid and energy metabolism would be depleted in a carbohydrate/amino acid/energy-rich environment like the rhizosphere. This is not an initially intuitive result for me.

Answer: We have revised it as below:

Specifically, the overrepresented metabolism pathways for universal rhizosphere depleted KOs are mainly involved in ATP synthesis, carbon fixation, amino acid biosynthesis,

nucleotides biosynthesis and other biosynthesis pathways (Fig 6G, Fig S26-S28), which might be due to that rhizosphere microbes can acquire diverse nutrients and ATP from plants.

332: How are the authors defining “positive selection” in this case? The analysis doesn’t seem to conform to analyses of natural selection from population genetics that I am familiar with.

Answer: We removed this part.

For your information, we used the same methods as described in reference 19. The protein sequences of each TIGRFAM family from the rhizosphere enriched or depleted MAGs were pooled and aligned using hmalign command of HMMER 3.1 (ref.97), then the alignments were trimmed using Trimal102 with parameter $-gt\ 0.5$, and the corresponding codon alignment was constructed using pal2nal 14 (ref.103) with default parameters based on the protein sequence alignment and the associated nucleotide sequences. Then a neighbor joining phylogenetic tree was constructed for each protein family using a modified clearcut19,104. The neighbor joining tree and the aligned codon sequences of each family from the rhizosphere enriched or depleted group were fed to HYPHY2.2 package105 and the Fast Unconstrained Bayesian AppRoximation (FUBAR)106 method was used to identify the positive selection affected TIGRFAM families with a threshold posterior probability >0.90 .

334: Please define ARGs.

Answer: We removed the comparison among locations, due to the reason described above.

335-338: It is possible that this enrichment is due to enhanced antibiotics used by these two countries but this could also be a function of anti-biotic production from other microbes or fungi

within the soils from those environments. If the authors have meta-data or a study that supports the claim that Brazil and China add exogenous antibiotics to combat common citrus disease, please cite.

Answer: We removed the comparison among locations, due to the reason described above.

Discussion

340: change “recruit” to “enrich”. Also, “huge” is a bit of an overstatement considering only 50+ genera make up the core of the community.

Answer: We revised it. Thanks.

340-355: The authors readily list the limitations in other studies and approaches but fail to acknowledge the limitations of their own. See above comments.

Answer: Thank you for your suggestions. We acknowledged the limitation of metagenomic study as below in the discussion: Another limitation of the metagenomic data is that it could not reveal whether the genes detected or enriched are expressed or functional at the tested conditions.

356-359: The authors only seem to recognize that metagenome sequencing has occurred in mammals and oceans while ignoring previous work that has been done in other soil systems. If they believe that this study is a better, or more in-depth, metagenomic analysis, please state. The contrasting of the rhizosphere with these environments is a bit of a straw-man argument as they are enriching for differences while ignoring previous work that has been done in

similar soil environments. Please make that comparison rather than these disparate studies that appear to have little to no relationship to the topic under discussion.

Answer: We revised it. Thanks.

356-379: This paragraph reads as more results rather than shaping and contextualizing the results into a cohesive narrative with a broader perspective to literature. Consider moving portions to the results.

Answer: We revised it. Thanks.

383-387: This conclusion is misleading as the depth of sequencing of DNA libraries for the metagenomic analysis was considerably deeper than the 16S sequencing. As shown in the rarefaction curves, both methods are still under sampling the extreme diversity present in these samples.

Answer: Thanks. We removed this conclusion.

393-397: Personally, I question how many of these are “true” novel genomes that exist in nature and how many of these are simply artifacts of bioinformatic genome construction. Additionally, the authors should address how intra-specific and intra-generic genetic variation can affect the bioinformatic construction of hard to culture microbes. Just because a computer can generate it, doesn’t mean it is real.

Answer: Thank you for your suggestions. We removed this part.

420: the data “suggest”, they do not “indicate”.

Answer: We revised it. Thanks.

422: How do these data “...underline the common mechanism used by plants to recruit microbes...”? The study only used a single plant genus and does a poor job comparing to the rhizosphere studies in other plants.

Answer: Thanks. We revised it as below:

“Our data suggest that both microbe-microbe interactions (synergistic, beneficial, and antagonistic) and plant recruitment play critical roles in the citrus-associated microbiome assembly. Those data underline the common mechanism used by citrus to recruit microbes and will provide the useful knowledge for the plant-associated synthetic microbiome engineering in citrus as well as in other plants.

424-417: Do not bring the topic back to the larger comparisons of human and ocean microbiomes that they set out to explain in the first place. This conclusion is far too narrow given the attempted scope of the study.

Answer: We revised it. Thanks.

Reviewers' comments:

Reviewer #1 (Remarks to the Author):

This review concerns the revised manuscript by Xu et al. describing structure and function of the citrus root and rhizosphere microbiomes. Although I recognize that numerous of my previous concerns were implemented, I have concerns related to the newly integrated metagenome comparisons (desert vs. non-desert soils, free-living vs. host-associated). The main worry is that the presented metagenome comparisons could be confounded by differences in sampling intensities and I suspect that normalization is needed for meaningful comparisons. Then, the re-assessment of the manuscript accentuated the need of making the article more accessible to a broad readership. It currently reads very technical with limited biological interpretation of the findings. The manuscript contains very many graphs and tables but it feels as the distillation of the data remains to the reader.

Major comments:

Technical language: The manuscript has a strong technical character and what is often lacking is a flavor of biology. For instance, the abstract advertises the large-scale gene catalogue from the six continents together with the deep sequencing and how many unigenes were captured, but remains short with what was the motivation or hypothesis and what has been learned from this study. The abstract does not include the core taxa and traits that were defined. Also, it remains unclear how this work advances our understanding of plant root-associated microbiomes and how it should become useful for synthetic microbiome engineering. Similarly, what has been learned from the conducted comparisons among microbiomes? In general, it would be beneficial to "guide" the reader through the biology, e.g. explaining why the comparison to other soils or to other biomes were conducted. For instance, explain the rationale why comparing the soil microbiomes of citrus plantations to the desert soil and non-desert soils. What was the motivation or hypothesis and what has been learned from this comparison?

Vocabulary: refrain from using expressions such as "plant root-associated soil microbiome" (does this expression refer to the rhizosphere?) or "citrus-associated bulk soil". The rhizosphere is considered as plant- or citrus-associated but "bulk soil" cannot be associated to a plant. Although the authors sampled soil from a citrus plantation, this soil is - sensu strictu sensu strictu - not citrus-associated as other plants can also grow in this soil. There are further opportunities to uniform the vocabulary of the manuscript: There is not needed to spell "16S rRNA gene V4" throughout the manuscript, technical jargon is sufficient in the methods. Consistent with the term "metagenomic data" would be "amplicon data".

Manuscript structure: My previous comment on manuscript structure remains partly unsolved. The abstract still presents first function and then structural information or the results sections starts with the sequencing effort of the metagenome data followed by the "taxonomic content...". Again some panels in Fig. 5 were miss-labeled.

Normalization: For several comparisons, I am concerned that they are meaningful. For instance, the numbers of detected genera (Fig. S5B). While the amplicon data has a sampling depth of 2M sequences (corresponding to $\frac{2}{183}$ number of OTUs) the counter part was sampled at a depth of 183M unigenes. The numbers reported in L154-155 strongly reflect the differences in sampling depth and are therefore not much informative. The deeper you sequence, the more you will find... I think some sort of a normalization is required here. One could for instance compare the genera that have >1% relative abundance (without considering the low abundant taxa). Similarly, the numeric comparison of genera presence and enrichment primarily reflects the differences in sequencing depths between the metagenomic and amplicon data (L191 and ff). A fair comparison between the compartments cannot be drawn.

Network analysis (L207 and ff.): the conducted network analysis does not permit to assess potential relationships among rhizosphere and root taxa. It is unclear, how Fig. 4 would have been prepared from only the core-rhizosphere data. Fig. 4 reveals the co-abundance behavior of bacteria genera in soil and rhizosphere samples, the ones on the left side of the hashed line are co-abundant in soil, the ones on the right side co-abundant in rhizosphere samples. The negative correlations between left and right are simply because the abundant soil taxa are low abundant in the rhizosphere and vice versa. Any comparison of distinct microbial habitats would reveal such type of a graph. What is the novelty of such a finding? Fig. S11 shows the same for the soil to root comparison. The two network analyses do not permit any assessment among rhizosphere and root taxa.

Other plant species: Just do confirm one thing related to the comparison with other plant species: were these bacteria community profiles obtained using the same PCR primers and 16S rRNA gene regions or might the presented variation be confounded by different profiling approaches? What is the rational/ statistic approach to determine abundant and dominant phyla and genera?

Desert vs. non-desert soils: Explain the rational of choosing these desert and non-desert soil microbiomes. How many distinct sites or soil types were assessed in these microbiomes? The global sampling of the authors accounts for 20 distinct sites while the two other types of out-group microbiomes have 7-9 samples. Hence, it appears plausible that a higher number of identified genera (L251) is found for the soil microbiomes of the 20 global citrus plantations. Furthermore, the number of identified genera will strongly depend on sampling depth. What are the sampling depths of the desert and non-desert soil microbiomes? I am again concerned related to normalization. Are the numbers of detected genera between soil from citrus plantation, desert and non-desert soils (Fig. 5B, but presented in panel C!) derived from data that was normalized by sampling depth? I think that biologically meaningful comparisons require normalization.

Free-living vs. host-associated: Explain the rational or hypothesis for this comparison. The claim related to microbiome complexity (L272) will strongly depend on sampling depths of the ocean and gut microbiomes. I am again worried about normalization as I think that biologically meaningful comparisons require even sampling depths. The way the results section is written, it lists the functional traits for the different investigated biomes, but does not really compare free-living with host-associated. In the discussion, the authors state the 3 times higher size of the citrus soil and rhizosphere metagenome compared to the other metagenomes, the enhanced sampling depth does not demonstrate that the soil is more complex (L362). The deeper you sequence, the more you will find. A complexity claim must be based on normalized data (something like nr of unigenes per unit of sampling depth).

Minor comments:

- General: The manuscript figures are highly colorful and to avoid confusion, it would be helpful to have the same color coding throughout the MS (e.g., the phylum Proteobacteria are in some graphs light pink, in others orange and yet in others in different green tones).
- L69: Why introducing the topic of adaptation of local microbes to introduced host species, this was not experimentally studied here.
- L130: The low levels of fungi, protozoa and plants should be mentioned in the discussion. Can this be related to the DNA extraction method? More general, the manuscript focusses strongly on the bacteria and while many analyses of the 16S rRNA gene amplicon data they were not conducted with the ITS amplicon data. For instance DESeq 2 analysis genus and OTU level, was not conducted.
- Fig 1D: Do not over-plot soil and rhizosphere samples. I do not see, what is the contribution of the Fig. S3.
- L146: What does "integratively" mean – compare side-by-side
- L146: Are the metagenomic unigenes splitted for taxonomy and function? E.g. are the 16S or ITS gene counts extracted for the comparison with the amplicon data?
- Fig. S9B: OTU-IDs missing, where is corresponding fungal analysis?
- The presented experimental design and data analysis does not permit any conclusions related to

agricultural practices (L245).

- Fig. S16: Why is there no further separation between the soil from citrus plantations and human and ocean samples? What type of dissimilarity index was used and I think, NMDS or PCoA would be more appropriate for the type of data.
- Fig. S20: I kind of have the same question as for Fig. S16, while the PC1 summarizes 87% of variation between citrus soil and the non/desert soil, PC1 accounts for only 49% between human, soil and ocean. I have no explanation for this.
- L403: The root compartment was not functionally assessed.

Reviewer #2 (Remarks to the Author):

The manuscript "The structure and function of the global citrus root-associated microbiome" by Xu, et. al. has improved from the initial draft. The authors have made a concerted effort to appropriately temper the initial overly-ambitious claims, which this reviewer appreciates. The current manuscript represents a considerable body of additional work but continues to suffer from a major issues with organization and presentation. This severely limits the understanding and communication of the analysis and hinders this reader's ability to understand the central message of the paper.

My primary concerns are as follows:

- Define the central message of the paper. The manuscript has several "hanging" analyses that muddle the message, including the root endosphere work, comparison with unrelated environments, and comparisons with other plant species with a total of 28 supplementary figures. Obviously, not all this work contributes to a cohesive message and the authors need to define what the central focus of the paper is and stick to it. From my perspective, the central focus is the comparison of 16S and metagenomic analysis to look at citrus rhizospheres vs bulk soil. Everything else should be considered for removal and placed into another paper. Especially with regard to the work on the citrus root endosphere, which would be better represented in a separate manuscript.
- Restructure the results. As written, the manuscript jumps from metagenomics to 16S and then back again. The results would be more cogent by starting with the 16S analysis describing what taxa are present and enriched in then rhizosphere and then moving on to the metagenomics describing what those taxa are functionally doing in the soil vs the rhizosphere.
- Lack of attention to grammatical detail. Several formatting/attributional mistakes. Several errors in citation formats as well as the references themselves. I have included many detailed comments listed below in the hope that it will improve the manuscript.
- Remove analysis on line 222-240. While I appreciate the attempt at an empirical analysis, the rhizosphere enrichment will be highly dependent on the soil used in the study and thus, direct comparisons cannot be mathematically compared in a statistically robust manner. This should be relegated to qualitative discussion with a sentence or two in the discussion.
- Many of the comparisons with other studies are not statistically controlled for the study. The authors need to statistically control for this (e.g. through constrained ordinations) to avoid confounding results with study conditions. For example, in Fig S20, the first PC accounts for 87.5% of the variance and appears to just separate out the studies.

Detailed concerns:

Figures:

Heatmap: The pastel blue-to-red color scheme is hard to read, doesn't communicate significance, and is not accessible for individuals with red/green colorblindness.

Figures S4 and S5 are not cogently organized and are confusing to follow. Please split up and re-organize.

Abstract:

47: remove "...biogeochemical cycles and". It is beyond the scope of the paper.

48-49: remove sentence "However, the genomic...." This work is focused on citrus, not plants in general.

The abstract needs a concluding sentence. It is incomplete as written.

Introduction:

64-69: Extremely awkward introductory paragraph. Both the subject and predicate of each sentence jump around and there is no attention to the flow of ideas from one sentence to the next. Also, these lines should be it's own paragraph. Please revise.

66: Also, needs citation for rhizosphere microbes helping with nutrient absorption in non-nitrogen-fixing and mycorrhizal-associated plants, especially for citrus.

70: Change "Modulation..." to "Optimization..."

71: "...them..." vague pronoun. Please revise.

76: Change to "...most previous studies..."

78-80: What is meant by "sufficient information"? Sufficient for what end? Please consider removing.

89: format of citation 35 is incorrect.

90: Remove "globally" as an adverb. It is vague and could reference "across the globe" or "extensively through the microcosm". Consider using "... bulk soil samples from across the globe, ..." at the end of the clause to clarify.

94: remove the words "rRNA gene V4". It is excessive detail except for the methods and interferes with the readability of the sentence.

100: Change "The previous study..." to "A previous study..." as the work does not appear to be a study from the ICM consortium.

106: Change "...the foundation..." to "...a foundation...". This is not the end-all, be-all references for all plant microbe interactions.

Results:

117-118: remove the "..., but not root microbiome..." There is no need to list what you did not do.

126-130: Please mention that this is expected based on 16S studies that show that there is much less taxonomic diversity in human guts and oceans (e.g. Thompson, 2017).

140: remove "rRNA gene V4"

142: remove "V4"

143-144: Please add the appropriate digit separators for the journal (commas I believe).

149-152: Please separate prokaryotes and fungi for a general audience. Most readers may not be able to identify which phyla are prokaryote vs eukaryote.

153: Error in Fig. S4. Glomeromycota should be with fungal sequences, not prokaryotes. Also, please remove root endosphere from the analysis since there is no metagenomic analysis to compare and is beyond the focus of the paper.

154: Don't start a sentence with number.

154-157: It is not fair to compare the 16S and metagenomic data in this fashion as there is a HUGE difference in sequencing depth.

155: Please add the appropriate digit separators for the journal (commas I believe).

161: Figs. S5C&D and S6: the analysis at the Genera level is too much detail and doesn't add much to story. Please remove.

165: remove "(Shannon index)" from the manuscript. Label in the figure, but in the text, it just breaks up the flow of the sentence.

164-166: This statement is not true for U.S. soils. See next.

166: this is an inappropriate test for what the authors are attempting to conclude, especially considering that HSD is not designed for unbalanced experimental designs. They should do a pair-wise ANOVA with microcosm (Bulk soil vs Rhizosphere) and site (AUS, BR, CN, etc). The graph would imply that changes in diversity are highly dependent on location. For example, AUS has no enrichment, BR has reduced rhizosphere enrichment, and the US looks like it may have higher rhizosphere enrichment. This can be followed by a post-hoc t-tests within site. Multiple test correction is not necessary for so few sites (i.e. the chances of a false positive due to multiple tests is low for less than 10 tests at a nominal p-value of 0.05). This analysis is extremely important as it shows that rhizosphere enrichment is highly dependent on the soil.

168: How was "significantly contributed" determined. It should be assessed with a two-way PERMANOVA. Please show the resulting F-table from the analysis.

175: Fig S8-S10: Can't tell what is significant. Please change all non-significant interactions to grey. Also, changes colorscheme as pastel colors are hard to follow. Consider, saturated blue and yellow as they are colorblind friendly.

189: Please state in the discussion that the criteria for "core citrus rhizosphere" may also be selecting for microbial populations that are simply common to soils where citrus can grow. The study can't separate out plant-associated microbes that are common soils where citrus grows from citrus specific taxa. In other words, would we expect the same "core citrus rhizosphere" for weeds growing the same plot.

203: "...which are the potential plant beneficial microbes." Awkward construction. Please revise. Maybe a separate statement.

212-213: Please provide an enrichment test (e.g. hypergeometric test) to support this statement. Can be relative to an Erdos-Renyi random network model.

222-240: These results are quantitatively comparable or appropriate. Please remove.

255: Fig S14: The ordination is extremely skewed by study collection. Please re-do with a constrained ordination (CA, DCA, or CCA) controlling for study. Also, Fig S20A

265 and 267: Figs 5B and 5C are miss attributed.

278: Fig16B, the citrus in the ordination do not appear distinct from the human or ocean samples and cannot be used as evidence of "...three different niches" as stated. Also, niche is mis-used in this context. These are three completely separate environments. Please remove analysis.

280: Again, please mention that this would be expected based on 16S taxonomic studies, such as Thompson 2017.

297: "...were absent in the two other niches". This is a hanging clause and I'm curious why the authors think this may be the case given that some of these KOs are common to all life (e.g. genetic information processing). Is this simply due to the depth of sequencing within a sample rather than sequencing more samples shallowly? Please expand. Also, remember niche is mis-used in this context. A niche is the position or role of an individual taxa within an environment while while an environment is the collection of abiotic conditions that shape a microbial community.

298-299: This implies that study is likely confounding the results. Especially for KOs common to all life, such as "genetic information processing".

321: Change "for" to "as". The sentence is confusing.

332: "as well as" is awkward in this instance. Please revise.

Discussion

344: The phrase "Different from the relatively simple environment of the human and animal gut, ..." is a terrible way to start a discussion as the gut is not a major focus of the paper. Remove.

246-347: change "...with the majority of which are based on ..." to "...using..." for clarity and brevity.

249: The phrase "Noteworthy, recent application ..." is awkward and I am not sure what the authors mean.

351-357: Several grammatical mistakes including spaces and commas. Please read and revise carefully. Also, consider starting a new sentence at 357 after "microbiomes".

393-395: Define modular microbiomes. Change "which" to "that". Also, potentially awkward construction.

305-397: Remove this sentence. Not germane to the discussion.

402-407: Long sentence. Break up to clarify the message.

407-409: Please support this statement. I'm not sure what data from your study you are using to make this claim.

419-422: Please support this statement. I'm not sure what data from your study you are using to make this claim.

424-426: End sentence after "...data sets." for clarity.

433: split sentence after "...complex."

Response to reviewers' comments

Reviewer #1 (Remarks to the Author):

This review concerns the revised manuscript by Xu et al. describing structure and function of the citrus root and rhizosphere microbiomes. Although I recognize that numerous of my previous concerns were implemented, I have concerns related to the newly integrated metagenome comparisons (desert vs. non-desert soils, free-living vs. host-associated). The main worry is that the presented metagenome comparisons could be confounded by differences in sampling intensities and I suspect that normalization is needed for meaningful comparisons. Then, the re-assessment of the manuscript accentuated the need of making the article more accessible to a broad readership. It currently reads very technical with limited biological interpretation of the findings. The manuscript contains very many graphs and tables but it feels as the distillation of the data remains to the reader.

Major comments:

Technical language: The manuscript has a strong technical character and what is often lacking is a flavor of biology. For instance, the abstract advertises the large-scale gene catalogue from the six continents together with the deep sequencing and how many unigenes were captured, but remains short with what was the motivation or hypothesis and what has been learned from this study. The abstract does not include the core taxa and traits that were defined. Also, it remains unclear how this work advances our understanding of plant root-associated microbiomes and how it should become useful for synthetic microbiome engineering. Similarly, what has been learned from the conducted comparisons among microbiomes? In general, it would be beneficial to “guide” the reader through the biology, e.g, explaining why the comparison to other soils or to other biomes were conducted. For instance, explain the rationale why comparing the soil microbiomes of citrus plantations to the desert soil and non-desert soils. What was the motivation or hypothesis and what has been learned from this comparison?

Response: Thank you for your critical but constructive suggestions. We have conducted further analysis and rewritten the manuscript. We believe the current manuscript is less technical, but focuses on the biology of the citrus rhizosphere microbiome.

Vocabulary: refrain from using expressions such as “plant root-associated soil microbiome” (does this expression refer to the rhizosphere?) or “citrus-associated bulk soil”. The rhizosphere is considered as plant- or citrus-associated but “bulk soil” cannot be associated to a plant. Although the authors sampled soil from a citrus plantation, this soil is - sensu strictu sensu strictu - not citrus-associated as other plants can also grow in this soil. There are further opportunities to uniform the vocabulary of the manuscript: There is not needed to spell “16S rRNA gene V4” throughout the manuscript, technical jargon is sufficient in the methods. Consistent with the term “metagenomic data” would be “amplicon data”.

Response: Thank you. We revised “plant root-associated soil microbiome” to “rhizosphere microbiome”; “16S rRNA gene V4” to “amplicon data”.

Manuscript structure: My previous comment on manuscript structure remains partly unsolved. The

abstract still presents first function and then structural information or the results sections starts with the sequencing effort of the metagenome data followed by the “taxonomic content...”. Again some panels in Fig. 5 were miss-labeled.

Response: Thank you. We re-organized the manuscript structure as you suggested. In the current version, we firstly showed the results of basic amplicon and shotgun metagenomic sequencing data, then presented the “Taxonomic content in global citrus rhizosphere and the associated soil microbiomes”, defined the “Core taxa of the citrus rhizosphere microbiome”; next, we reported “Core functional traits in global citrus rhizosphere microbiome”. According to this manuscript structure, we re-organized all the figures and tables.

Normalization: For several comparisons, I am concerned that they are meaningful. For instance, the numbers of detected genera (Fig. S5B). While the amplicon data has a sampling depth of 2M sequences (corresponding to $\frac{2}{183}$ number of OTUs) the counter part was sampled at a depth of 183M unigenes. The numbers reported in L154-155 strongly reflect the differences in sampling depth and are therefore not much informative. The deeper you sequence, the more you will find... I think some sort of a normalization is required here. One could for instance compare the genera that have >1% relative abundance (without considering the low abundant taxa). Similarly, the numeric comparison of genera presence and enrichment primarily reflects the differences in sequencing depths between the metagenomic and amplicon data (L191 and ff). A fair comparison between the compartments cannot be drawn.

Response: Thank you. The comparison between amplicon and metagenome data, and citrus root microbiome data are out of our focus, we removed this part in current version. In the future, we will further focus on these questions.

Network analysis (L207 and ff.): the conducted network analysis does not permit to assess potential relationships among rhizosphere and root taxa. It is unclear, how Fig. 4 would have been prepared from only the core-rhizosphere data. Fig. 4 reveals the co-abundance behavior of bacteria genera in soil and rhizosphere samples, the ones on the left side of the hashed line are co-abundant in soil, the ones on the right side co-abundant in rhizosphere samples. The negative correlations between left and right are simply because the abundant soil taxa are low abundant in the rhizosphere and vice versa. Any comparison of distinct microbial habitats would reveal such type of a graph. What is the novelty of such a finding? Fig. S11 shows the same for the soil to root comparison. The two network analyses do not permit any assessment among rhizosphere and root taxa.

Response: Agree. We removed this part in current version.

Other plant species: Just do confirm one thing related to the comparison with other plant species: were these bacteria community profiles obtained using the same PCR primers and 16S rRNA gene regions or might the presented variation be confounded by different profiling approaches? What is the rational/statistic approach to determine abundant and dominant phyla and genera?

Response: Thank you. We removed this part in current version.

Desert vs. non-desert soils: Explain the rationale of choosing these desert and non-desert soil microbiomes. How many distinct sites or soil types were assessed in these microbiomes? The global sampling of the authors accounts for 20 distinct sites while the two other types of out-group microbiomes have 7-9 samples. Hence, it appears plausible that a higher number of identified genera (L251) is found for the soil microbiomes of the 20 global citrus plantations. Furthermore, the number of identified genera will strongly depend on sampling depth. What are the sampling depths of the desert and non-desert soil microbiomes? I am again concerned related to normalization. Are the numbers of detected genera between soil from citrus plantation, desert and non-desert soils (Fig. 5B, but presented in panel C!) derived from data that was normalized by sampling depth? I think that biologically meaningful comparisons require normalization.

Response: Thank you. We removed these contents in current version. In the future, we will focus on these questions using proper methods as you suggested.

Free-living vs. host-associated: Explain the rationale or hypothesis for this comparison. The claim related to microbiome complexity (L272) will strongly depend on sampling depths of the ocean and gut microbiomes. I am again worried about normalization as I think that biologically meaningful comparisons require even sampling depths. The way the results section is written, it lists the functional traits for the different investigated biomes, but does not really compare free-living with host-associated. In the discussion, the authors state the 3 times higher size of the citrus soil and rhizosphere metagenome compared to the other metagenomes, the enhanced sampling depth does not demonstrate that the soil is more complex (L362). The deeper you sequence, the more you will find. A complexity claim must be based on normalized data (something like nr of unigenes per unit of sampling depth).

Response: Thank you. We removed these contents in current version. In the future, we will focus on these questions using proper methods as you suggested.

Minor comments:

- General: The manuscript figures are highly colorful and to avoid confusion, it would be helpful to have the same color coding throughout the MS (e.g., the phylum Proteobacteria are in some graphs light pink, in others orange and yet in others in different green tones).

Response: Agree. We revised the colors as you suggested.

- L69: Why introducing the topic of adaptation of local microbes to introduced host species, this was not experimentally studied here.

Response: Agree. We removed this sentence.

- L130: The low levels of fungi, protozoa and plants should be mentioned in the discussion. Can this be related to the DNA extraction method? More general, the manuscript focusses strongly on the bacteria and while many analyses of the 16S rRNA gene amplicon data they were not conducted with the ITS amplicon data. For instance DESeq 2 analysis genus and OTU level, was not conducted.

Response:

1. We mentioned the possible reason for the low levels of fungi, protozoa and plants in our data set: As in earlier studies that used a metagenomic approach to profile the plant-associated microbiome, we found that bacteria dominated the rhizosphere and that eukaryotes accounted for a small fraction of the sequences that could be associated with known taxa. The apparently low proportional representation of eukaryotes in the rhizosphere probably results from the fact that the our taxonomic classification method to identify community composition was reference-based, and the reference genomes of most eukaryotes are not available. Such a conjecture is supported by the fact that more than 59% of the unigenes could not be assigned to any known taxon.
2. For the DNA extraction method, we used MoBio Powersoil DNA extraction kit, which was the standard method for soil microbiome samples in Earth Microbiome Project.
3. We also conducted the analyses of ITS amplicon data, such as DESeq2 analysis for phylum and genus level. See the supplementary table 2, 3. Due to a few significant different genera for ITS amplicon data, we did not describe this data using heatmap. For the metagenomics data, we mentioned the eukaryotic genus in Fig 3b and Figs S6-S7. And we also added the ITS results in current version.

• Fig 1D: Do not over-plot soil and rhizosphere samples. I do not see, what is the contribution of the Fig. S3.

Response: Agree. We removed Fig S3.

• L146: What does “integratively” mean – compare side-by-side

Response: We have changed it to: Here, we determined the identity of microbes in the citrus rhizosphere and associated bulk soil primarily by metagenomic sequencing, but complemented such assessments using amplicon sequences.

• L146: Are the metagenomic unigenes splitted for taxonomy and function? E.g. are the 16S or ITS gene counts extracted for the comparison with the amplicon data?

Response: Yes, we conducted the taxonomic and functional annotation using all the unigenes based on nr with the MEGAN LCA algorithm and KEGG database, respectively. Because the assembly for the 16S or ITS gene was poor, we did not extract these genes from metagenomic data to perform the taxonomic annotation.

• Fig. S9B: OTU-IDs missing, where is corresponding fungal analysis?

Response: We removed the OTUs comparison part. We conducted the taxonomic (phylum and genus), and functional comparisons between rhizosphere and bulk soil mainly based on metagenomic data.

• The presented experimental design and data analysis does not permit any conclusions related to agricultural practices (L245).

Response: Agree. We removed it.

- Fig. S16: Why is there no further separation between the soil from citrus plantations and human and ocean samples? What type of dissimilarity index was used and I think, NMDS or PCoA would be more appropriate for the type of data.

Response: We removed these contents in current version.

- Fig. S20: I kind of have the same question as for Fig. S16, while the PC1 summarizes 87% of variation between citrus soil and the non/desert soil, PC1 accounts for only 49% between human, soil and ocean. I have no explanation for this.

Response: We removed these contents in current version.

- L403: The root compartment was not functionally assessed.

Response: We removed the root compartment to focus on citrus rhizosphere because we do not have data for roots from all the locations.

Reviewer #2 (Remarks to the Author):

The manuscript “The structure and function of the global citrus root-associated microbiome” by Xu, et. al. has improved from the initial draft. The authors have made a concerted effort to appropriately temper the initial overly-ambitious claims, which this reviewer appreciates. The current manuscript represents a considerable body of additional work but continues to suffer from a major issues with organization and presentation. This severely limits the understanding and communication of the analysis and hinders this reader’s ability to understand the central message of the paper.

My primary concerns are as follows:

- Define the central message of the paper. The manuscript has several “hanging” analyses that muddle the message, including the root endosphere work, comparison with unrelated environments, and comparisons with other plant species with a total of 28 supplementary figures. Obviously, not all this work contributes to a cohesive message and the authors need to define what the central focus of the paper is and stick to it. From my perspective, the central focus is the comparison of 16S and metagenomic analysis to look at citrus rhizospheres vs bulk soil. Everything else should be considered for removal and placed into another paper. Especially with regard to the work on the citrus root endosphere, which would be better represented in a separate manuscript.

Response: Agree. As you suggested, we removed the comparison between citrus microbiome data and other data sets, such as human gut, ocean, desert and non-desert soil and other plant root-associated microbiomes, and the citrus root microbiome data. In current version, we focus on the structure and function of global citrus rhizosphere microbiome.

- Restructure the results. As written, the manuscript jumps from metagenomics to 16S and then back

again. The results would be more cogent by starting with the 16S analysis describing what taxa are present and enriched in then rhizosphere and then moving on to the metagenomics describing what those taxa are functionally doing in the soil vs the rhizosphere.

Response: Agree, thank you. We re-organized the manuscript structure as you suggested. We used both amplicon and metagenomics data for the taxonomic annotation. So, in the current version, we firstly showed the results of basic amplicon and shotgun metagenomic sequencing data, then presented the “Taxonomic content in global citrus rhizosphere and the associated soil microbiomes”, defined the “Core taxa of the citrus rhizosphere microbiome”; next, we reported “Core functional traits in global citrus rhizosphere microbiome”.

- Lack of attention to grammatical detail. Several formatting/attributional mistakes. Several errors in citation formats as well as the references themselves. I have included many detailed comments listed below in the hope that it will improve the manuscript.

Response: To deal with grammatical detail, both Drs. Steven E. Lindow and Frank White have helped us revise the manuscript.

- Remove analysis on line 222-240. While I appreciate the attempt at an empirical analysis, the rhizosphere enrichment will be highly dependent on the soil used in the study and thus, direct comparisons cannot be mathematically compared in a statistically robust manner. This should be relegated to qualitative discussion with a sentence or two in the discussion.

Response: Agree. We removed this part in current version.

- Many of the comparisons with other studies are not statistically controlled for the study. The authors need to statistically control for this (e.g. through constrained ordinations) to avoid confounding results with study conditions. For example, in Fig S20, the first PC accounts for 87.5% of the variance and appears to just separate out the studies.

Response: Agree. We removed these contents in current version.

Detailed concerns:

Figures:

Heatmap: The pastel blue-to-red color scheme is hard to read, doesn't communicate significance, and is not accessible for individuals with red/green colorblindness.

Response: We changed the color to blue/yellow for the heatmap.

Figures S4 and S5 are not cogently organized and are confusing to follow. Please split up and re-organize.

Response: We re-organized these figure. See new Fig S3 and S4.

Abstract:

47: remove "...biogeochemical cycles and". It is beyond the scope of the paper.

Response: We removed it.

48-49: remove sentence "However, the genomic...." This work is focused on citrus, not plants in general. The abstract needs a concluding sentence. It is incomplete as written.

Response: Thank you for your good suggestions. We rewrote the abstract as follow.

Citrus is a globally important perennial fruit crop whose rhizosphere microbiome is believed to play an important role in promoting citrus growth and health. However, the genomic and functional components of the citrus rhizosphere microbiome remain largely unknown. To obtain a comprehensive understanding of the structural and functional compositions of the citrus rhizosphere microbiome, we performed both amplicon and deep shotgun metagenomic sequencing of bulk soil and rhizosphere samples collected across distinct biogeographical regions from six continents. Several taxa, including Proteobacteria, Actinobacteria, Acidobacteria and Bacteroidetes, were predominant in the global citrus rhizosphere microbiome. Furthermore, we characterized the core citrus rhizosphere microbiome, comprised of Pseudomonas, Agrobacterium, Cupriavidus, Bradyrhizobium, Rhizobium, Mesorhizobium, Burkholderia, Cellvibrio, Sphingomonas, Variovorax and Paraburkholderia, some of which are potential plant beneficial microbes. We also examined those core citrus rhizosphere microbial functional traits that were over-represented in this habitat that mediated plant-microbe and microbe-microbe interactions, nutrition acquisition and plant growth promotion. The results provide valuable information to guide microbial isolation, and culturing, and to harness the power of the microbiome to improve plant production and health.

Introduction:

64-69: Extremely awkward introductory paragraph. Both the subject and predicate of each sentence jump around and there is no attention to the flow of ideas from one sentence to the next. Also, these lines should be it's own paragraph. Please revise.

Response: Thank you for your good suggestions. We rewrote the introduction.

66: Also, needs citation for rhizosphere microbes helping with nutrient absorption in non-nitrogen-fixing and mycorrhizal-associated plants, especially for citrus.

Response: In current version, we removed this sentence.

70: Change "Modulation..." to "Optimization..."

Response: We removed this sentence.

71: "...them..." vague pronoun. Please revise.

Response: We removed this sentence during revision.

76: Change to "...most previous studies..."

Response: We revised it as suggested.

78-80: What is meant by "sufficient information"? Sufficient for what end? Please consider removing.

Response: We removed it.

89: format of citation 35 is incorrect.

Response: We revised it.

90: Remove "globally" as an adverb. It is vague and could reference "across the globe" or "extensively through the microcosm". Consider using "... bulk soil samples from across the globe, ..." at the end of the clause to clarify.

Response: We revised it.

94: remove the words "rRNA gene V4". It is excessive detail except for the methods and interferes with the readability of the sentence.

Response: We removed it.

100: Change "The previous study..." to "A previous study..." as the work does not appear to be a study from the ICM consortium.

Response: Agree. We removed these contents in current version.

106: Change "...the foundation..." to "...a foundation...". This is not the end-all, be-all references for all plant microbe interactions.

Response: We revised it.

Results:

117-118: remove the "..., but not root microbiome..." There is no need to list what you did not do.

Response: We removed it during revision.

126-130: Please mention that this is expected based on 16S studies that show that there is much less taxonomic diversity in human guts and oceans (e.g. Thompson, 2017).

Response: Thank you for your good suggestions. We removed this content.

140: remove “rRNA gene V4”

Response: We removed it.

142: remove “V4”

Response: We removed it.

143-144: Please add the appropriate digit separators for the journal (commas I believe).

Response: We revised it.

After removal of sequences associated with the citrus host, on average 21,942 and 22,797 16S rDNA tags and 21,523 and 22,555 ITS2 tags were generated for each bulk soil and rhizosphere sample, respectively (Table S2 & Table S3).

149-152: Please separate prokaryotes and fungi for a general audience. Most readers may not be able to identify which phyla are prokaryote vs eukaryote.

Response: We revised it.

The dominant prokaryotic phyla found in the citrus rhizosphere included Proteobacteria, Actinobacteria, Acidobacteria and Bacteroidetes, while fungal phyla, included Ascomycota and Basidiomycota (Fig 2a, Fig S3 and Fig S4).

153: Error in Fig. S4. Glomeromycota should be with fungal sequences, not prokaryotes. Also, please remove root endosphere from the analysis since there is no metagenomic analysis to compare and is beyond the focus of the paper.

Response: Agree, thanks. We revised it. We described the eukaryotes and prokaryotes separately. Please see Fig S3 and S4.

154: Don't start a sentence with number.

Response: Agree, we removed it.

154-157: It is not fair to compare the 16S and metagenomic data in this fashion as there is a HUGE difference in sequencing depth.

Response: Agree, we removed it.

155: Please add the appropriate digit separators for the journal (commas I believe).

Response: We removed it.

161: Figs. S5C&D and S6: the analysis at the Genera level is too much detail and doesn't add much to story. Please remove.

Response: We removed it.

165: remove "(Shannon index)" from the manuscript. Label in the figure, but in the text, it just breaks up the flow of the sentence.

Response: Agree. We removed it.

164-166: This statement is not true for U.S. soils. See next.

166: this is an inappropriate test for what the authors are attempting to conclude, especially considering that HSD is not designed for unbalanced experimental designs. They should do a pair-wise ANOVA with microcosm (Bulk soil vs Rhizosphere) and site (AUS, BR, CN, etc). The graph would imply that changes in diversity are highly dependent on location. For example, AUS has no enrichment, BR has reduced rhizosphere enrichment, and the US looks like it may have higher rhizosphere enrichment. This can be followed by a post-hoc t-tests within site. Multiple test correction is not necessary for so few sites (i.e. the chances of a false positive due to multiple tests is low for less than 10 tests at a nominal p-value of 0.05). This analysis is extremely important as it shows that rhizosphere enrichment is highly dependent on the soil.

Response: Agree. We have conducted the analyses per your suggestions.

168: How was "significantly contributed" determined. It should be assessed with a two-way PERMANOVA. Please show the resulting F-table from the analysis.

Response: Agree, thanks. We re-analyzed this as you suggested.

Principal coordinate analysis (PCoA) and variation partitioning analysis (VPA) based on unweighted UniFrac distance (beta diversity) also revealed that the community composition of the rhizosphere and bulk soil did not differ (P-value >0.05, F-value =1.23 using permutation-based ANOVA, Fig 2c and Fig S5b).

175: Fig S8-S10: Can't tell what is significant. Please change all non-significant interactions to grey. Also, changes colorscheme as pastel colors are hard to follow. Consider, saturated blue and yellow as they are colorblind friendly.

Response: Thank you for your good suggestions. We changed the color to blue/yellow for the heatmap.

189: Please state in the discussion that the criteria for "core citrus rhizosphere" may also be selecting for microbial populations that are simply common to soils where citrus can grow. The study can't separate out plant-associated microbes that are common soils where citrus grows from citrus specific taxa. In other words, would we expect the same "core citrus rhizosphere" for weeds growing the same plot.

Response: We added this in the discussion as follow.

Although we defined the core rhizosphere microbiome under the aforementioned criteria, some of these core rhizosphere microbes may be common in all soils where citrus is planted but may not be specific for citrus. Consequently, further experiments are needed to define the specific core citrus rhizosphere microbiome.

203: "...which are the potential plant beneficial microbes." Awkward construction. Please revise. Maybe a separate statement.

Response: Thank you for your good suggestions. We revised it.

The core rhizosphere microbes, such as *Pseudomonas*, *Agrobacterium*, *Cupriavidus*, *Bradyrhizobium*, *Rhizobium*, *Shinella*, *Mesorhizobium*, *Burkholderia*, *Cellvibrio*, *Sphingomonas*, *Variovorax*, *Paraburkholderia*, *Dyadobacter*, *Novosphingobium*, *Devosia* and *Ensifer*, were over-represented in *Proteobacteria* (corrected P-value <0.05, Fisher's exact test) (Fig 3b, Fig S7a). Multiple members affiliated with these core bacterial genera are known as plant beneficial microbes, and these microbes might help maintain plant hormone balance, control root development, facilitate nutrition acquisition, and prevent disease in the plant host⁴⁰⁻⁴².

212-213: Please provide an enrichment test (e.g. hypergeometric test) to support this statement. Can be relative to an Erdos-Renyi random network model.

Response: Per the other reviewer's suggestions, we removed this content.

222-240: These results are quantitatively comparable or appropriate. Please remove.

Response: Agree. We removed it.

255: Fig S14: The ordination is extremely skewed by study collection. Please re-do with a constrained ordination (CA, DCA, or CCA) controlling for study. Also, Fig S20A

Response: We removed these contents in current version.

265 and 267: Figs 5B and 5C are miss attributed.

Response: We removed these contents in current version.

278: Fig16B, the citrus in the ordination do not appear distinct from the human or ocean samples and cannot be used a evidence of "...three different niches" as stated. Also, niche is mis-used in this context. These are three completely separate environments. Please remove analysis.

Response: We removed these contents in current version.

280: Again, please mention that this would be expected based on 16S taxonomic studies, such as Thompson 2017.

Response: We removed these contents in current version.

297: "...were absent in the two other niches". This is a hanging clause and I'm curious why the authors think this may be the case given that some of these KOs are common to all life (e.g. genetic information processing). Is this simply due to the depth of sequencing within a sample rather than sequencing more samples shallowly? Please expand. Also, remember niche is mis-used in this context. A niche is the position or role of an individual taxa within an environment while an environment is the collection of abiotic conditions that shape a microbial community.

Response: We removed these contents in current version.

298-299: This implies that study is likely confounding the results. Especially for KOs common to all life, such as "genetic information processing".

Response: We removed these contents in current version.

321: Change "for" to "as". The sentence is confusing.

Response: Agree. We revised it.

332: "as well as" is awkward in this instance. Please revise.

Response: Thank you for your good suggestions. We revised it.

These core functional traits were mainly involved in plant-microbe and microbe-microbe interactions and pathways that might be anticipated to nutrient acquisition of microbes. The rhizosphere-depleted functional traits were involved in genetic information processing and metabolic pathways, such as carbohydrate metabolism, amino acid biosynthesis, energy metabolism and nucleic acid biosynthesis (Fig 4e, Fig S8 and Supplementary data 4).

Discussion

344: The phrase "Different from the relatively simple environment of the human and animal gut, ..." is a terrible way to start a discussion as the gut is not a major focus of the paper. Remove.

Response: We removed it.

246-347: change "...with the majority of which are based on ..." to "...using..." for clarity and brevity.

Response: Agree. We revised it.

Previous studies of rhizosphere and soil microbiomes have been mainly based on amplicon sequencing approaches^{3-7,9,10,12,13,16,17,19,20,55,56}.

249: The phrase “Noteworthy, recent application ...” is awkward and I am not sure what the authors mean.

Response: We removed it.

351-357: Several grammatical mistakes including spaces and commas. Please read and revise carefully. Also, consider starting a new sentence at 357 after “microbiomes”.

Response: We removed it because it is not essential.

393-395: Define modular microbiomes. Change “which” to “that”. Also, potentially awkward construction.

Response: We removed it during revision.

305-397: Remove this sentence. Not germane to the discussion.

Response: Removed.

402-407: Long sentence. Break up to clarify the message.

Response: We revised it.

407-409: Please support this statement. I’m not sure what data from your study you are using to make this claim.

Response: We removed this sentence and rewrote this part.

419-422: Please support this statement. I’m not sure what data from your study you are using to make this claim.

Response: We removed these contents in current version.

424-426: End sentence after “...data sets.” for clarity.

Response: We removed these contents in current version.

433: split sentence after “...complex.”

Response: We removed these contents in current version.

REVIEWERS' COMMENTS:

Reviewer #3 (Remarks to the Author):

Review of "The Structure and function of the global citrus rhizosphere microbiome" by Xu, et al, submitted to Nature Communications.

In this manuscript, Xu and Colleagues collect rhizosphere soil from citrus trees around the world, and bulk reference soils for most of the trees, and conduct microbiome analysis and metagenomes.

The manuscript provides a large scale analysis of the citrus microbiome from several continents, which is unique. The paper is generally well-written.

The results themselves are fairly unremarkable. The value in the paper seems to be that these samples are from global locations and for an agronomically important crop. However, this is a data-intensive manuscript that doesn't really grow our knowledge about plant-microbe associations.

My take home from the paper is that 1) the citrus rhizosphere looks like the rhizosphere of other plants and 2) that the metagenomic information confirms this and shows that "KOs involved in known plant-microbe and microbe-microbe interactions (line 218)" were enriched. We've known this about the rhizosphere for a while and these findings do not substantially advance our knowledge of the importance of microbiomes in general. Even more specifically, these results tell us nothing about the potential function of the citrus-specific microbiome.

Several things that would have made the paper stronger: A discussion of the differences in communities by location. There are clearly geographic differences in both fig s2 and Fig 2c that are interesting and probably important. Likewise, there should be a comparison of how/whether the geography is more important or not to the microbial communities than whether the communities were derived from the rhizosphere or not.

Many of the analyses note phylum-level changes in composition, which makes poor use of the intense sampling effort. This is particularly an interesting choice of analyses considering the authors chose to use UPARSE, which clusters OTUs into ESVs - a method that generally gives more taxon-level information. There was also no discussion about taxa that couldn't be classified to genera level. Are these important? If so, how much. In my experience, only a small fraction of the OTUs can be assigned to the OTU level.

The biological interpretations are superficial and quite weak throughout the manuscript. I don't have any suggestions here for how to make that stronger. However, as I was reading the paper I had the thought that perhaps the enrichment of certain functions might be lined to the actual taxonomy of the enriched OTUS. Something to consider.

I struggle to understand the importance of identifying the "core" citrus microbiome. What does it do for the plant in specific terms? Is it different than other plants that would share the same soil? What is the role fo geography in the core citrus microbiome? I'm just struggling to understand this point.

Reviewer #3 (Remarks to the Author):

Review of “The Structure and function of the global citrus rhizosphere microbiome” by Xu, et al, submitted to Nature Communications.

In this manuscript, Xu and Colleagues collect rhizosphere soil from citrus trees around the world, and bulk reference soils for most of the trees, and conduct microbiome analysis and metagenomes.

The manuscript provides a large scale analysis of the citrus microbiome from several continents, which is unique. The paper is generally well-written.

Response: Thank you so much. We appreciate your comments.

The results themselves are fairly unremarkable. The value in the paper seems to be that these samples are from global locations and for an agronomically important crop. However, this is a data-intense manuscript that doesn't really grow our knowledge about plant-microbe associations.

Response: Thanks for your comments. In this manuscript, we report a comprehensive analysis of the structural and functional compositions of the citrus rhizosphere microbiome using both amplicon and deep shotgun metagenomic sequencing of bulk soil and rhizosphere samples collected across distinct biogeographical regions from six continents. Based on this large scale data information, we defined the core citrus microbiomes and their functional traits. The results provide valuable information to guide microbial isolation and culturing and, potentially, to harness the power of the microbiome to improve plant production and health.

My take home from the paper is that 1) the citrus rhizosphere looks like the rhizosphere of other plants and 2) that the metagenomic information confirms this and shows that “KOs involved in known plant-microbe and microbe-microbe interactions (line 218)” were enriched. We've known this about the rhizosphere for a while and these findings do not substantially advance our knowledge of the importance of microbiomes in general. Even more specifically, these results tell us nothing about the potential function of the citrus-specific microbiome.

Response: Thanks for your comments. There are more take home messages besides you mentioned. Most studies of plant-associated microbial communities have been conducted by means of ribosomal amplicon-based approaches. However, amplicon-based community profiling does not provide either the genomic or functional details of the microbiome. In addition, the global pattern of the genomic and functional contents of rhizosphere microbial communities remains largely unexplored. In this study, we conducted a comprehensive analysis of the structural and functional compositions of the citrus rhizosphere microbiome. We use both amplicon and deep shotgun metagenomic sequencing of bulk soil and rhizosphere samples collected across distinct biogeographical regions from six continents. Predominant taxa include *Proteobacteria*, *Actinobacteria*, *Acidobacteria* and *Bacteroidetes*. The core citrus rhizosphere microbiome comprises *Pseudomonas*, *Agrobacterium*, *Cupriavidus*, *Bradyrhizobium*, *Rhizobium*, *Mesorhizobium*, *Burkholderia*, *Cellvibrio*, *Sphingomonas*, *Variovorax* and *Paraburkholderia*, some of which are potential plant beneficial microbes. We also identify over-represented

microbial functional traits mediating plant-microbe and microbe-microbe interactions, nutrition acquisition and plant growth promotion in citrus rhizosphere. This study lay a foundation for harnessing the microbiome for sustainable citrus production.

Several things that would have made the paper stronger: A discussion of the differences in communities by location. There are clearly geographic differences in both fig s2 and Fig 2c that are interesting and probably important. Likewise, there should be a comparison of how/whether the geography is more important or not to the microbial communities than whether the communities were derived from the rhizosphere or not.

Response: Thank you for your constructive suggestions. Some previous studies (Bulgarelli et al., 2012; Lundberg et al., 2012) have confirmed that location is the most important factor to shape the soil and rhizosphere microbiome communities. In our study, we focus on the core citrus rhizosphere microbiome and their functional traits across distinct biogeographical regions from six continents.

Many of the analyses note phylum-level changes in composition, which makes poor use of the intense sampling effort. This is particularly an interesting choice of analyses considering the authors chose to use UPARSE, which clusters OTUs into ESVs - a method that generally gives more taxon-level information. There was also no discussion about taxa that couldn't be classified to genera level. Are these important? If so, how much. In my experience, only a small fraction of the OTUs can be assigned to the OTU level.

Response: Thanks for your suggestions. Amplicon-based community composition analysis is a classical approach for microbiome analysis. However, shotgun metagenomic sequences generated without PCR amplification can also be used to determine the identity and relative abundance of microbes whose presence might not be detected in ribosomal gene amplicons due to primer bias and have been successfully utilized in interrogation of diverse microbiomes. Here, we determined the identity of microbes in the citrus rhizosphere and associated bulk soil primarily by metagenomic sequencing, but complemented such assessments using amplicon sequences. Furthermore, the metagenomic sequences obtained provided more comprehensive taxonomic information and given the community compositions made by this method were consistent with that from the amplicon sequences, this method was chosen to define a core rhizosphere microbiome.

The majority of the microbes in soil are unknown and uncultivable. Due to the limited references information, it is very common that many taxa that couldn't be classified to genera level and OTU level. We have also discussed the limitation for metagenomics on the eukaryotes as following:

As in earlier studies that used a metagenomic approach to profile the plant-associated microbiome, we found that bacteria dominated the rhizosphere and that eukaryotes accounted for a small fraction of the sequences that could be associated with known taxa. The apparently low proportional representation of eukaryotes in the rhizosphere probably results from the fact that the our taxonomic classification method to identify community composition was reference-based, and the reference genomes of most eukaryotes are not available. Such a conjecture is

supported by the fact that more than 59% of the unigenes could not be assigned to any known taxon.

The biological interpretations are superficial and quite weak throughout the manuscript. I don't have any suggestions here for how to make that stronger. However, as I was reading the paper I had the thought that perhaps the enrichment of certain functions might be lined to the actual taxonomy of the enriched OTUS. Something to consider.

Response: Thank you for suggestions. We respectfully disagree. We have done a thorough interpretation of the global citrus rhizosphere microbiome and have found many interesting information. For example, predominant taxa include Proteobacteria, Actinobacteria, Acidobacteria and Bacteroidetes in rhizosphere microbiome. The core citrus rhizosphere microbiome comprises *Pseudomonas*, *Agrobacterium*, *Cupriavidus*, *Bradyrhizobium*, *Rhizobium*, *Mesorhizobium*, *Burkholderia*, *Cellvibrio*, *Sphingomonas*, *Variovorax* and *Paraburkholderia*, some of which are potential plant beneficial microbes. We found functions related to carbohydrate metabolism and amino acid metabolism are under-represented in the rhizosphere core microbiome. This would suggest that the resources and microenvironment provided by plants does not differ much between plant species. The rhizosphere enrichment of bacterial secretion systems, chemotaxis, flagella, assembly, nutrient transporters, antimicrobial resistance and antibiotic synthesis genes indicates that the coevolution of host-microbe and microbe-microbe interactions can be logically linked to the conditions present in the rhizosphere, thus accounting for their positive selection. It is therefore expected that rhizosphere enrichment of transcriptional factors would also be associated with such microbes enriched in the rhizosphere because they would be required for proper expression of adaptations to this habitat. Interestingly, some CRISPR-associated proteins were enriched in the bulk soil microbiome, indicating that microbes face more intense selection pressures from bacteriophages. Phage infection might be expected to be more prominent in rhizosphere environments due to their higher population sizes, allowing epidemics of viral infection to occur. Consistent with the identification of potential plant beneficial microbes in the citrus rhizosphere, the core functional traits of the citrus rhizosphere microbiome are likely involved in enhancing nutrient uptake by plants as well as modulating hormonal balances, thereby influencing environmental adaptation and the prevention of pathogenic infection in plants. This observation supports that core rhizosphere microbes provide benefits to plant growth and health.

I struggle to understand the importance of identifying the “core” citrus microbiome. What does it do for the plant in specific terms? Is it different than other plants that would share the same soil? What is the role of geography in the core citrus microbiome? I'm just struggling to understand this point.

Response: Busby et al. 2017 have done an excellent job in addressing the importance of core plant microbiome. They recommended that the research priorities for harnessing plant microbiomes for sustainable agriculture include determining the functional mechanisms mediating plant-microbiome interactions and defining the core microbiome of crop and non-crop plant species (Busby et al. 2017).

Busby, P. E. et al. Research priorities for harnessing plant microbiomes in sustainable agriculture. *PLOS Biol.* 15, e2001793 (2017)